# SIMO v1.0: simplified model of the vertical temperature profile in a small, warm, monomictic lake

**Kristina Šarović**[1]**, Melita Burić**[2]**, and Zvjezdana B. Klaić**[1]

[1]Department of Geophysics, Faculty of Science, University of Zagreb, Zagreb, 10000, Croatia
[2]Gekom Geophys & Ecol Modelling Ltd, Zagreb, 10000, Croatia

**Correspondence:** Kristina Šarović (kristina.sarovic@gmail.com)

**Abstract.** A simple 1-D energy budget model (SIMO) for the prediction of the vertical temperature profiles in small, monomictic lakes forced by a reduced number of input meteorological variables is proposed. The model estimates the net heat flux and thermal diffusion using only routinely measured hourly mean meteorological variables (namely, the air temperature, relative humidity, atmospheric pressure, wind speed, and precipitation), hourly mean ultraviolet B radiation (UVB), and climatological yearly mean temperature data. Except for the initial vertical temperature profile, the model does not use any lake-specific variables. The model performance was evaluated against lake temperatures measured continuously during an observational campaign in two lakes belonging to the Plitvice Lakes, Croatia (Lake 1 and Lake 12). Temperatures were measured at 15 and 16 depths ranging from 0.2 to 27 m in Lake 1 (maximum depth of 37.4 m) and 0.2 to 43 m in Lake 12 (maximum depth of 46 m). The model performance was evaluated for simulation lengths from 1 to 30 d. The model performed reasonably well, and it was able to satisfactorily reproduce the vertical temperature profile at the hourly scale, the deepening of the thermocline with time, and the annual variation in the vertical temperature profile, which shows its applicability for short-term prognostic simulations. A yearlong simulation initiated with an approximately constant vertical profile of the lake temperature ($\approx 4\,°\mathrm{C}$) was able to reproduce the onset of stratification and convective overturn. The epilimnion temperature was somewhat overestimated, especially with the onset of the convective overturn. The upper limit of the metalimnion was well captured, while its thickness was overestimated. Nevertheless, the values of the model performance measures obtained for a yearlong simulation were comparable with those reported for other, more complex models. Thus, the presented model can also be used for long-term simulations and the assessment of the onset and duration of lake stratification periods when water temperature data are unavailable, which can be useful for various lake studies performed in other scientific fields, such as biology, geochemistry, and sedimentology.

## 1 Introduction

Water temperature is a critical factor that directly influences a whole range of lake properties. It controls the solubility of gases and minerals, the rate of chemical reactions, and biological activity and diversity (e.g., Benson and Krause, 1980; Rasconi et al., 2017; Krumgalz, 2018). In addition, the vertical temperature profile in a lake (and consequent lake stratification and water column stability) and the length of the stratification period play a vital role in the transport pathways of gases and nutrients and, consequently, their distribution within a lake (e.g., Vachon et al., 2019; Ladwig et al., 2021). Furthermore, there is a two-way interaction between lakes and the atmosphere. While the thermodynamic behavior of lakes is mainly driven by meteorological conditions, the distinct physical features of lakes (such as surface roughness, albedo, heat capacity and/or temperature, and evaporation rate) introduce surface heterogeneity in the domain of interest. Thus, their presence modifies surface–atmosphere fluxes and local and regional weather and climate (e.g., MacKay, 2012; Klaić and Kvakić, 2014; Bryan et al., 2015; Kristovich et al., 2017; Wu et al., 2019). Thus, over the last couple of decades, increasing scientific interest has been focused on

both modeling the thermal regime of lakes (e.g., Stepanenko et al., 2013, 2016; Thiery et al., 2014; MacKay et al., 2017) and its sensitivity to climate change (e.g., Råman Vinnå et al., 2021).

Due to their relative simplicity and computational efficiency, there is a widespread use of one-dimensional (1-D) water temperature prediction models. There are different types of 1-D models of varying complexity, although they can generally be divided into three groups: (1) mixed layer models based on the energy budget approach, (2) differential models based on solving the 1-D heat transfer equation (thermal diffusivity models), and (3) second-order turbulence closure models. Energy budget-based models assume series of well-mixed layers (often just two, namely, the epilimnion and hypolimnion), and they use the kinetic energy produced by wind shear on the surface to account for the mixing dynamics within these layers and/or to estimate their depths (e.g., Bell et al., 2006; Mironov et al., 2010; Hipsey et al., 2019). Thermal diffusivity-based models usually consist of many well-mixed layers for which the heat transfer equation is solved (e.g., Hostetler and Bartlein, 1990; Liston and Hall, 1995; Stefan et al., 1998; Sun et al., 2007). The second-order turbulence closure models are also known as $k$-$\varepsilon$, where $k$ is the turbulent kinetic energy per unit mass and $\varepsilon$ is the turbulent kinetic energy dissipation rate (e.g., Goudsmit et al., 2002). They solve the turbulent kinetic energy transport equation and are, computationally, considerably more expensive than the previous two types (e.g., Goudsmit et al., 2002; Stepanenko et al., 2011, 2014).

Except in the basic underlying approach, lake models differ in the processes they include, such as wind sheltering, sediment heat flux, attenuation of light, phase change, convective mixing, and others. Direct implementation of a particular process in a model or the simplification or even omission of the process is usually justified by the model purpose. Lake models are developed for various purposes, including improvement of numerical weather prediction and climate models (e.g., Mironov et al, 2010; MacKay, 2012), evaluation of the effects of climate change (e.g., Stefan et al., 1998; Wu et al., 2020; Råman Vinnå et al., 2021), or facilitation of specific limnological studies. Some of these specific studies address gas (e.g., methane and/or $CO_2$) emissions (e.g., Stepanenko et al., 2011), oxygen and nutrient levels (e.g., Bell et al., 2006), heat and mass exchange between the atmosphere and a water body (Sun et al., 2007), and evaporation and lake level fluctuation (Hostetler and Bartlein, 1990).

To run lake models, input data, which are generally not available from routine meteorological measurements, are needed. Specifically, these data include both shortwave and longwave radiation component data. The goal of this study is to formulate a simplified model for predicting the vertical temperature profile in a small, warm, monomictic lake, which, except for the ultraviolet B radiation (UVB), is forced solely by routinely available observed surface meteorological data (namely, the air temperature, relative humidity, atmo-

spheric pressure, wind speed, and precipitation). Conversely, other lake temperature models that are forced with observational data (e.g., Bell et al., 2006; Sun et al., 2007; Martynov et al., 2010; MacKay, 2012, 2017) require both shortwave and longwave radiation component data and do not provide further details on determining them. The proposed model employs carefully chosen parameterizations of longwave and shortwave radiation. Although these parametrizations are well known, in the present study, they are built into a lake temperature model for the first time. Furthermore, in comparison with the model of Sun et al. (2007), the present model does not neglect the turbulent diffusion for small lakes and uses a different approach for calculating the light attenuation with depth. In addition, we examined the sensitivity of the proposed model performance to the length of the simulated time interval. To the best of our knowledge, such a detailed evaluation has not been reported in previous lake temperature modeling studies. Since vertical temperature profiles in lakes are not routinely measured, we also addressed the ability of the proposed model to simulate the onset and termination of lake stratification by a yearlong simulation initiated with a uniform temperature over a completely mixed water column. A similar study was performed by Martynov et al. (2010) for two small dimictic lakes in the USA using an eddy diffusivity model and a two-layer model; Goudsmit et al. (2002) analyzed the performance of a $k$-$\varepsilon$ model in a two-year length simulation, while Bruce et al. (2018) analyzed a set of 32 lakes all over the globe using the General Lake Model (GLM). All of these models are more complex and/or require more extensive input data than the one proposed in this study.

The model proposed here is evaluated using lake temperature experimental data measured at two lakes of the Plitvice Lakes, Croatia. Details about the study area and data collection are presented in Sect. 2. The model's governing equations and parametrizations used are described in Sect. 3. Measures of the model performance and evaluation approach are described in Sect. 4. The results are presented and discussed in Sect. 5, and a comparison with other models is presented in Sect. 6. Finally, a short summary and conclusions are given in Sect. 7.

## 2 Study area and measurements

### 2.1 Study area

Plitvice Lakes is a karstic lake system situated in the mountainous region of Croatia (Fig. 1). The system consists of 16 named and several smaller unnamed lakes. The lakes are interconnected with cascades and waterfalls, making a chain approximately 9 km long and extending in a roughly south–north direction. With its unique geomorphology and exceptional biodiversity, the area has been a subject of scientific research dating as early as 1850 (NPPL, 2021). An exten-

sive, multidisciplinary overview of abiotic studies focusing on the Plitvice Lakes area is provided by Klaić et al. (2018).

The numerical model proposed in this paper was applied to the two largest lakes of the system, Prošće and Kozjak Lakes (Fig. 1c and d). Prošće Lake (hereafter Lake 1) is the southernmost and the first lake in the system, while Kozjak Lake (hereafter Lake 12) is the 12th lake in the chain and the largest and deepest lake in the system. The characteristics of each lake are given in Table 1. Based on their surface areas, both lakes can be considered small (e.g., Forcat et al., 2011).

## 2.2 Observational data

### 2.2.1 Lake temperatures

This study uses lake temperatures measured at two different points (Fig. 1), one in Lake 1 (point P1, $\varphi = 44.8676°$ N, $\lambda = 15.5981°$ E, 636 m a.s.l.) and the other in Lake 12 (point K1, $\varphi = 44.8902°$ N, $\lambda = 15.6038°$ E, 535 m a.s.l.). Each point was positioned in the deepest part of the corresponding lake. Lake temperatures were measured and logged with HOBO TidBiT MX Temp 400, as previously described for Lake 1 in Klaić et al. (2020b) and for Lake 12 in Klaić et al. (2020a). The accuracy of the sensors is $\pm 0.20\,°$C for temperatures between 0 and 70 °C and $\pm 0.25\,°$C for temperatures between $-20$ and 0 °C. The initial sampling frequency of lake temperatures was 1 Hz, while 2 min means were stored. However, since meteorological data were available at a resolution of one hour, we used hourly mean lake temperatures in the present study.

At site P1 (Lake 1), 15 factory-calibrated sensors were positioned at fixed depths of 0.2, 0.5, 1, 1.5, 3, 5, 7, 9, 11, 13, 15, 17, 20, 23, and 27 m. As Lake 12 is deeper than Lake 1, an additional sensor was placed at a depth of 43 m at site K1 together with 15 sensors at the same depths as at site P1.

The temperature recording started on 7 July 2018 at K1 and 6 July 2019 at P1 (Table 2). Temperatures were recorded continuously, except during several short periods ($\approx 1$–2 d, once in approximately four months) when the sensors were pulled out of the lakes for the purpose of data acquisition. These periods without measurements are shown as thin, vertical white lines in Fig. 2a, c, and e. Due to the malfunction of some sensors during the first year of the measurement campaign, data for some observational depths at K1 are missing.

Missing data are shown as white areas from July 2018 to July 2019 in Fig. 2c or as intermitted lines in Fig. 2d. The inoperative sensors were later replaced. Missing data at specific depths were subsequently replaced by data calculated by spatial linear interpolation from the two adjacent depths using existing data (Fig. 2e and f). However, temporal interpolation was not performed, since it would fail to reproduce the temporal variability in lake temperature at particular depths during periods of data acquisition. Interpolated lake temperatures were used solely to illustrate the evolution of Lake 12

stratification (Fig. 2e); they were omitted in the calculations of the model performance measures (Sect. 4).

### 2.2.2 Meteorological data

Meteorological data were measured at the automatic meteorological station Plitvička Jezera (point M in Fig. 1, $\varphi = 44.8811°$ N, $\lambda = 15.6197°$ E, altitude 579 m a.s.l.). The station belongs to the network of the Croatian Meteorological and Hydrological Service (CMHS). The CMHS also provided quality control of these data. In the present study, we used hourly mean values of the air temperature, atmospheric pressure, UVB radiation, atmospheric relative humidity, hourly precipitation amount measured at 2 m above ground level, and wind speed measured at 10 m above ground level (Fig. 3). Wind direction data were also available but were not used in the study.

The station is approximately 2 km northeastward of the P1 site and 1.6 km southeastward of the K1 site. Despite the comparable distance from both the P1 and K1 sites, the meteorological conditions observed at point M are expected to be more representative for Lake 12 than for Lake 1, because this point is located at the slope adjacent to Lake 12 at approximately 200 m away from its shoreline. In addition, topographic obstacles are found between points P1 and M (Fig. 1b), and the altitude difference between P1 and M is higher than the difference between K1 and M (Table 2).

## 3 Model description and governing equations

The model is based on the one-dimensional energy balance equation used in similar liquid water models (e.g., Hostetler and Bartlein, 1990; Liston and Hall, 1995; Sun et al., 2007). Because ice was not observed on the two lakes during the measurement campaign (Fig. 2a and c), ice formation was not addressed in the present study. Thus, a simplified approach using water temperature instead of enthalpy is used. Considering that, more often than not, the lake bathymetry is not available, in addition to our goal to keep the model as simple as possible and to limit the input data, it is assumed that the water body has a constant horizontal cross-sectional area (which can be of any shape). Thus, we come to the following equation: TS1

$$c_\mathrm{p}\rho\frac{\partial T}{\partial t} = \frac{\partial}{\partial z}\left\{[k_\mathrm{m} + k_\mathrm{t}]\frac{\partial T}{\partial z}\right\} - \frac{\partial \phi}{\partial z} + M_\mathrm{conv}, \qquad (1)$$

where $c_\mathrm{p}$ is the water specific heat capacity (J kg$^{-1}$ K$^{-1}$), $\rho$ is the water density (kg m$^{-3}$), $T$ is the water temperature (°C), $t$ is time (s), $z$ is depth (m), $k_\mathrm{m}$ and $k_\mathrm{t}$ are the molecular and turbulent thermal conductivity (W m$^{-1}$ K$^{-1}$), $\phi$ is the heat flux (W m$^{-2}$), and $M_\mathrm{conv}$ is the convective mixing term (W m$^{-3}$).

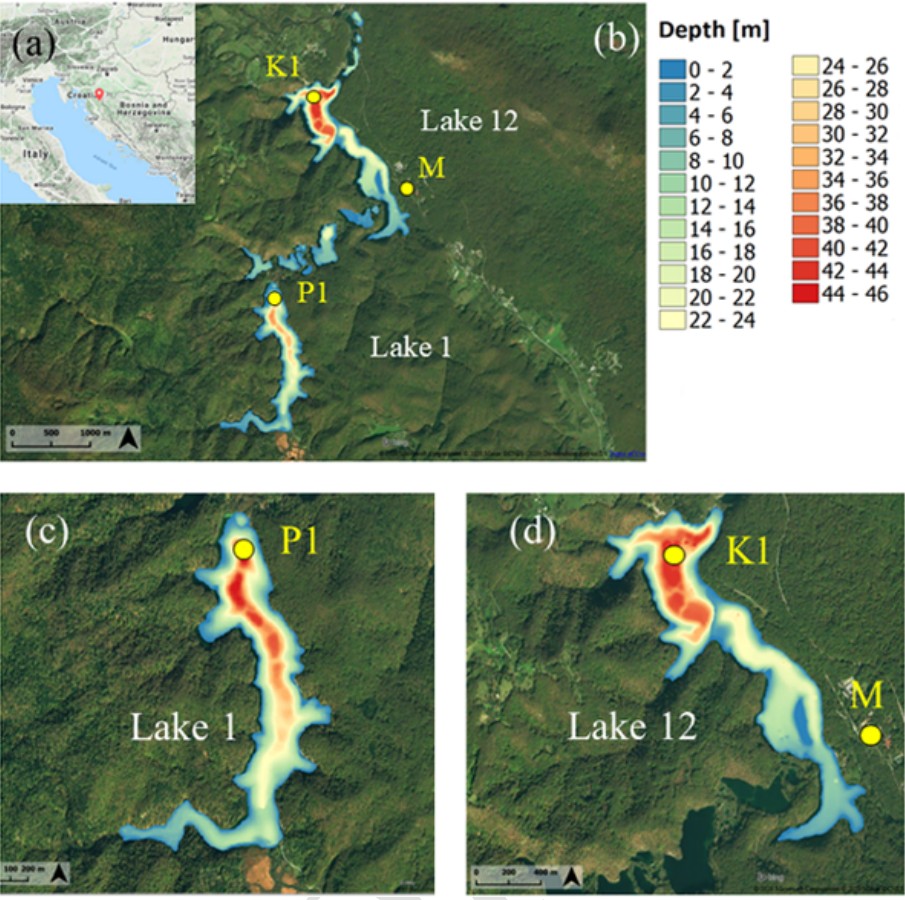

**Figure 1.** Location of Plitvice Lakes (red bubble; source: © Google Maps) **(a)**; closer look at the entire lake system **(b)**; Lake 1 **(c)**, and Lake 12 **(d)**. Locations of the lake temperature measuring points P1 ($\varphi = 44.8676°$ N, $\lambda = 15.5981°$ E, height of the lake surface 636 m a.s.l.) and K1 ($\varphi = 44.8902°$ N, $\lambda = 15.6038°$ E, 535 m a.s.l.), and meteorological measuring site M ($\varphi = 44.8811°$ N, $\lambda = 15.6197°$ E, 579 m a.s.l.) are shown with yellow circles. Panels **(b)**–**(d)** show composite pictures of the lake bathymetries and the digital orthophoto images (DOF: http://www.bing.com/maps/?v=2&app=60526, last access: 3 December 2020, Print Rights – Under the Microsoft® Bing™ Maps Platform APIs' Terms of Use).

**Table 1.** Characteristics of the studied lakes.

|  | Lake 1 (Prošće Lake) | Lake 12 (Kozjak Lake) |
|---|---|---|
| Altitude | 636.6 m a.s.l.* | 535 m a.s.l. |
| Area | 0.68 km² | 0.82 km² |
| Volume | 0.00767 km³ | 0.01271 km³ |
| Max. or Average depth | 37.4/13.2 m | 46/17.3 m |

* a.s.l. – above sea level.

The water density is calculated from the Chen and Millero (1986) formula, assuming zero salinity:

$$\rho = 999.8395 + 6.7914 \times 10^{-2}T - 9.0894 \times 10^{-3}T^2$$
$$+ 1.0171 \times 10^{-4}T^3 - 1.2846 \times 10^{-6}T^4 + 1.1592$$
$$\times 10^{-8}T^5 - 5.0125 \times 10^{-11}T^6. \quad (2)$$

The molecular thermal conductivity of water is 0.6 W m⁻¹ K⁻¹ (e.g., Sun et al., 2007). The turbulent thermal conductivity is a function of time and depth, because it depends on meteorological forcing. Here, we also follow the method of Henderson-Sellers (1985), where the turbulent thermal conductivity is calculated as follows: TS2

$$k_t(z) = c_p \rho \left(ku^*z/Pr_0\right) \exp\left(-k^*z\right) \left(1 + 37Ri^2\right)^{-1}, \quad (3)$$

where $k = 0.4$ is the von Karman constant, $u^*$ is the friction velocity at the surface (m s⁻¹), $k^*$ is the latitude-dependent

**Table 2.** Availability of measured data. The positions of the measuring points are shown in Fig. 1.

| Dataset | Measurement point | Availability of data |
|---|---|---|
| Water temperature | K1 (Lake 12, maximum depth 46 m) $\varphi = 44.8902°$ N, $\lambda = 15.6038°$ E, 535 m a.s.l. | 7 July 2018–2 November 2020 |
| Water temperature | P1 (Lake 1, maximum depth 37.4 m) $\varphi = 44.8676°$ N, $\lambda = 15.5981°$ E, 636 m a.s.l. | 6 July 2019–2 November 2020 |
| Meteorological data | M $\varphi = 44.8811°$ N, $\lambda = 15.6197°$ E, 579 m a.s.l. | 7 July 2018–4 November 2018 1 January 2019 – 31 December 2019 2 July 2020 – 30 September 2020 |

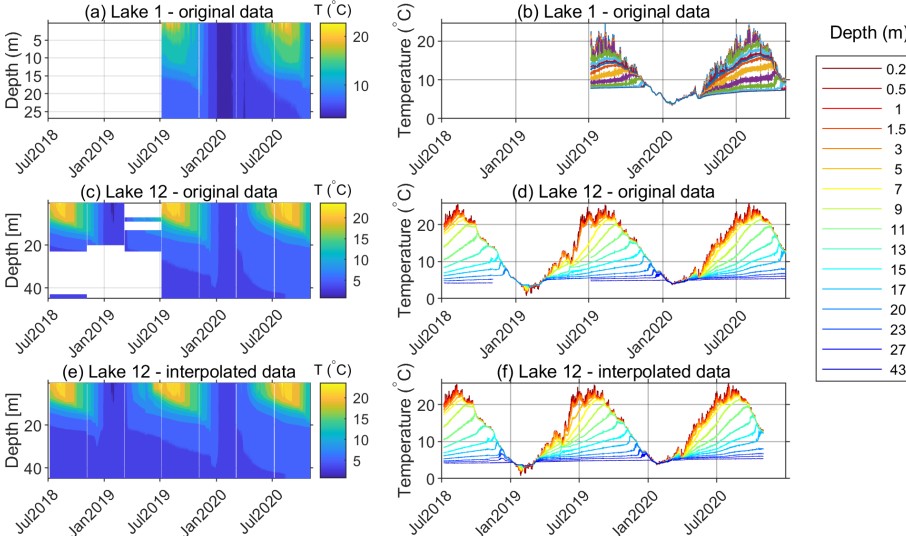

**Figure 2.** Measured water temperatures in Lake 1 (**a** and **b**) and Lake 12 (**c** and **d**) and water temperature in Lake 12 after interpolation of the measured data (**e** and **f**).

parameter of the Ekman profile, $Pr_0 = 1$ is the neutral value of the turbulent Prandtl number, and $Ri$ is the Richardson number. The Ekman profile parameter and the Richardson number are calculated as in Sun et al. (2007):

$$k^* = 6.6(\sin \phi)^{1/2} U_2^{-1.84}, \tag{4}$$

where $\varphi$ is the latitude and $U_2$ is the wind speed at 2 m above the water surface (m s$^{-1}$), and

$$Ri = \frac{-1 + \left\{ 1 + 40N^2 k^2 z^2 / \left[ u^{*2} \exp\left(-2k^* z\right) \right] \right\}^{1/2}}{20}, \tag{5}$$

where $N$ is the Brunt–Väisälä frequency (s$^{-1}$):

$$N = \left[ -g/\rho \left( \partial \rho / \partial z \right) \right]^{1/2}. \tag{6}$$

The wind speed $U_2$ is determined from the logarithmic formula:

$$U_2 = u^* \log\left(2/z_0\right) / k, \tag{7}$$

where $z_0$ is the roughness length (m). The air shear velocity $u^*$ and the roughness length $z_0$ are calculated as in Verburg and Antenucci (2010).

Although Sun et al. (2007) suggest that, for shallow lakes (less than 50 m deep), the turbulent thermal conductivity is negligible, this is not in accordance with findings of numerous other studies, which suggest that the turbulent thermal conductivity can be much larger than the molecular thermal conductivity, even for shallow lakes (e.g., Jassby and Powell, 1975; Quay et al., 1980; Vachon et al., 2019). It should be kept in mind that these studies often determine the turbulent diffusion coefficient based on measured change rate of lake water temperature vertical distribution. This means that the contributions of all present mixing processes are included (i.e., shear-induced turbulence, breaking internal waves, boundary layer turbulence). However, the mixing processes and their contributions to turbulent mixing may differ from lake to lake. In the present study, turbulent thermal diffusion was taken into account using Eq. (3).

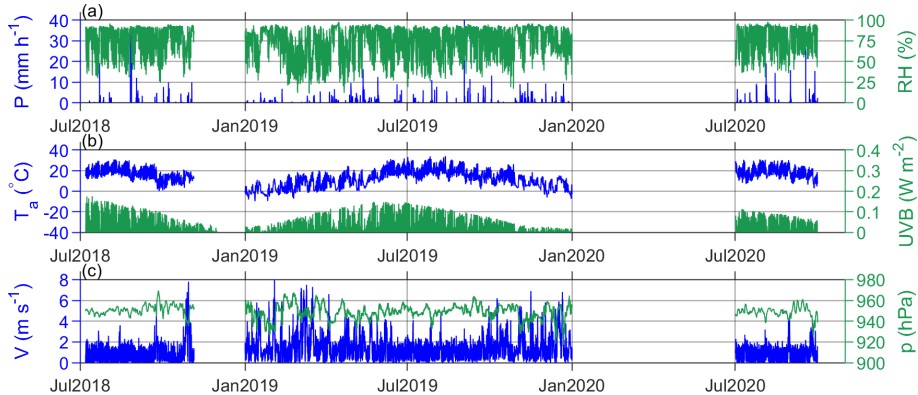

**Figure 3.** Available meteorological data from the automatic meteorological station Plitvička Jezera ($\varphi = 44.8811°$ N, $\lambda = 15.6197°$ E, 579 m a.s.l.): **(a)** precipitation amount ($P$) and relative humidity (RH), **(b)** air temperature ($T_a$) and UVB radiation (UVB), and **(c)** wind speed ($V$) and atmospheric pressure ($p$).

## 3.1 Energy budget and boundary conditions

In addition to turbulent thermal diffusion, the only other term in Eq. (1) accounting for meteorological forcing is the heat source term. The surface net heat flux consists of the net shortwave radiation ($S_n$), net longwave radiation ($L_n$), sensible heat flux ($H_s$), latent heat flux ($H_l$), and heat flux brought by precipitation ($H_p$). The surface boundary condition can be written as follows:

$$\phi(0) = S_n + L_n + H_s + H_l + H_p. \tag{8}$$

At the bottom, it is assumed that there is no heat flux and that the temperature gradient equals zero, meaning there is no heat diffusion either. Thus, the bottom boundary conditions can be written as follows:

$$\frac{\partial T}{\partial z}(z_{max}) = 0, \tag{9}$$

$$\phi(z_{max}) = 0. \tag{10}$$

All heat flux terms in Eq. (8) are defined to be positive when downward. Shortwave and longwave radiation measurements are not very common, and sensible and latent heat fluxes cannot be measured directly (Brunel, 1989; Bahr et al., 2012). Thus, obtaining the heat flux terms in Eq. (8) is expensive and complicated. Therefore, methods for calculating each term using commonly available meteorological data only are proposed in sections 3.1.1. to 3.1.4.

### 3.1.1 Shortwave radiation

As previously indicated by other authors (e.g., Bell et al., 2006; Martynov et al., 2010; MacKay, 2012), sufficient radiation data (both shortwave and longwave) are not generally available from routine meteorological measurements, and this is also the case for meteorological station M, where only UVB radiation was measured. A number of studies provide correlations among UVA, UVB, total UV, or global solar radiation ($G$) (Kudish and Evseev, 2000; Kudish et al.,

2005; Podstawczynska, 2009; Pokhrel and Bhattarai, 2012; Pashiardis et al., 2017) and show that significant variability occurs in the UV/$G$ ratio between sites, which is mainly due to local atmospheric conditions. Podstawczynska (2009) indicated that air turbidity and cloudiness are the two main factors that determine the variability of daily solar energy transmission through the atmosphere. Pashiardis et al. (2017) found that the UV/$G$ ratio increases with solar elevation and that the presence of clouds reduces the UV component less than the global solar radiation due to the strong absorption of water in the near-infrared spectrum.

Winslow et al. (2001) proposed a model for estimating the total daily solar irradiance from daily precipitation and minimum and maximum temperatures, along with latitude, elevation, and mean annual temperature. This model showed significant improvement over the widely used empirical Bristow and Campbell (1984) model and was applicable for a wide range of climates. Therefore, it is also used in this study.

According to Winslow et al. (2001), the daily solar irradiance at the Earth's surface is equal to

$$S_{surf} = \tau_{cf} D \left(1 - \beta_s \mathrm{rh}_{T_{max}}\right) S_{top}, \tag{11}$$

where TS3 $S_{surf}$ is the total daily solar irradiance at the surface (J m$^{-2}$ d$^{-1}$), $\tau_{cf}$ is the cloud-free atmospheric transmittance, $\beta_s$ is an additional parameter required to introduce variation between sites, $\mathrm{rh}_{T_{max}}$ is the relative humidity at the moment when daily maximum air temperature ($T_{max}$) is reached, and $S_{top}$ is the total daily solar irradiance at the top of the atmosphere (J m$^{-2}$ d$^{-1}$). The total daily solar irradiance is calculated following Wald (2019): TS4

$$S_{top} = S_0 (1 + \varepsilon_{ecc}) \frac{3600 \times 24}{\pi} \cos\phi \cos\delta \left[\omega_s - \tan(\omega_s)\right], \tag{12}$$

where $S_0 = 1362$ W m$^{-2}$ is the solar constant, $\varepsilon_{ecc}$ is the eccentricity of Earth's orbit, $\delta$ is the solar declination, $\varphi$ is the location latitude, and $\omega_s$ is the half-day length (time between

sunrise and noon or noon and sunset) in radians. $\varepsilon_{\mathrm{ecc}}$ and $\omega_{\mathrm{s}}$ are functions of the day in the year only, while $\delta$ also depends on the location longitude, since its noon value is used, which yields more precise results. Details on calculating these parameters are included in Wald (2019).

The cloud-free atmospheric transmittance in Eq. (11) accounts for the transmittance of dry clean air ($\tau_0$) and the transmittance due to absorption by aerosols ($\tau_a$) and water vapor ($\tau_v$), and it also incorporates a correction for elevation ($c_{\mathrm{elev}}$):

$$\tau_{\mathrm{cf}} = (\tau_0 \tau_a \tau_v)^{c_{\mathrm{elev}}}. \tag{13}$$

To calculate $\tau_0$, $\tau_a$, $\tau_v$, $c_{\mathrm{elev}}$, $D$, and $\beta_{\mathrm{s}}$, we follow Winslow et al. (2001). The transmittance of dry clean air is dependent only on the latitude ($\varphi$) and is calculated as follows:[TS5]

$$\tau_0 = 0.947 - \left(1.033 \times 10^{-5}\right)|\varphi|^{2.22} \text{ for } |\varphi| \le 80°$$

$$\tau_0 = 0.774 \text{ for } |\varphi| > 80°. \tag{14}$$

The absorption by aerosols is extremely variable. Similar to Winslow et al. (2001), we set $\tau_a = 1$ (i.e., no absorption).

The absorption by water vapor is calculated from the following:

$$\tau_v = 0.9636 - 9.092 \times 10^{-5}(T_{\mathrm{mean}} + 30)^{1.8232}, \tag{15}$$

where $T_{\mathrm{mean}}$ is the mean annual air temperature (°C). On wet days, when the daily precipitation is above 1 mm, $\tau_v$ is reduced by 0.13. The site elevation correction factor ($c_{\mathrm{elev}}$) is calculated as follows:

$$c_{\mathrm{elev}} = \left[1 - \left(2.2569 \times 10^{-5}\right) z_{\mathrm{a.s.l.}}\right]^{5.2553}, \tag{16}$$

where $z_{\mathrm{a.s.l.}}$ is the site elevation (m).

From Eq. (11), $\tau_{\mathrm{cf}} S_{\mathrm{top}}$ is the maximum cloud-free value of $S_{\mathrm{surf}}$. The effect of cloudiness is indirectly taken into account by introducing the factor $D(1 - \beta \mathrm{rh}_{T_{\mathrm{max}}})$. This is based on the finding that the solar irradiation from sunrise, when minimum humidity is expected ($\mathrm{rh}_{T_{\mathrm{min}}} \approx 1$), until the maximum daily air temperature (and minimum humidity $\mathrm{rh}_{T_{\mathrm{max}}}$) is reached, is proportional to the decline of the relative humidity, $S_{\mathrm{surf}\_T_{\mathrm{max}}} \propto (1 - \beta \mathrm{rh}_{T_{\mathrm{max}}})$. The factor $D = S_{\mathrm{surf}}/S_{\mathrm{surf}\_T_{\mathrm{max}}}$ is introduced to account for the surface solar irradiation from the moment when the air temperature reaches its daily maximum until sunset. $D$ is calculated assuming that the air temperature reaches its maximum around 3 pm:

$$D = \left[1 - (\omega_{\mathrm{s}} - \pi/4)^2 / \left(2\omega_{\mathrm{s}}^2\right)\right]^{-1}. \tag{17}$$

The factor $\beta_{\mathrm{s}}$ in Eq. (11) is mainly constant, except for regions with very large daily temperature ranges:

$$\beta_{\mathrm{s}} = \max\{1.041, 23.753 \Delta T_{\mathrm{m}}/(T_{\mathrm{mean}} + 273.16)\}, \tag{18}$$

where $\Delta T_{\mathrm{m}}$ is the mean annual temperature range between the daily air temperature maximum and minimum.

Hourly shortwave radiation data were generated from the calculated daily solar irradiance by using the measured UVB radiation data as a weight function:

$$S(h) = \mathrm{UVB}(h)\frac{S_{\mathrm{surf}}}{\mathrm{UVB}_{\mathrm{day}}} = \mathrm{UVB}(h)\frac{S_{\mathrm{surf}}}{3600\sum\limits_{h=1}^{24}\mathrm{UVB}(h)}, \tag{19}$$

where $S_{\mathrm{surf}}$ and [TS6]$\mathrm{UVB}_{\mathrm{day}}$ are the daily values ($\mathrm{J\,m^{-2}\,d^{-1}}$) and $S(h)$ and $\mathrm{UVB}(h)$ are the mean values ($\mathrm{W\,m^{-2}}$) for the $h^{th}$ hour of the total and UVB solar radiation, respectively. When UVB radiation data are unavailable, the standard daily radiation profile can be used.

Unlike the other terms in Eq. (8), shortwave radiation is not completely absorbed in the lake surface layer but partially passes through the water. The net shortwave radiation reaching a particular depth is calculated using the arctangent model, which was chosen for its simplicity for implementation, as suggested by Henderson-Sellers (1986), but also for its better representation of the light attenuation in the shallow layers, which are usually a lot thinner than the deeper ones:

$$S_{\mathrm{n}}(z) = (1 - \alpha)S\exp(-K_1 z)\left[1 - K_2\tan^{-1}(K_3 z)\right], \tag{20}$$

where $S_{\mathrm{n}}(z)$ is the net shortwave radiation at water depth $z$ ($\mathrm{W\,m^{-2}}$), $\alpha = 0.06$ is the water surface albedo, and $K_1$, $K_2$, and $K_3$ are empirical constants. $K_1$ corresponds to the light extinction coefficient $\lambda_{\mathrm{e}} = 0.1$ (value of 0.1 is appropriate for clear oligotrophic lakes). $K_2$ is calculated as

$$K_2 = 2\left[1 - (1 - \beta)\exp(\lambda_{\mathrm{e}} z_A)\right]/\pi, \tag{21}$$

where $\beta = 0.4$ accounts for the absorption in the surface layer, and $z_A = 0.6\,\mathrm{m}$ is the depth of the surface absorption layer, where the exponential decay starts. The third parameter, $K_3 = 4$, is not a direct function of $\lambda_{\mathrm{e}}$ and $\beta$ but is rather a measure of the rapidity of falloff with depth in the upper layers.

### 3.1.2 Longwave radiation

The net longwave radiation is the difference between the incoming downward atmospheric longwave radiation ($L_{\mathrm{a}}^{\downarrow}$) and the outgoing upwards radiation from the lake surface ($L_{\mathrm{s}}^{\uparrow}$). As direct measurement data of longwave radiation by pyrgeometers are not routinely available, longwave radiation may be calculated using the following formula:

$$L_{\mathrm{n}} = (1 - r)L_{\mathrm{a}}^{\downarrow} - L_{\mathrm{s}}^{\uparrow} = (1 - r)\left[\varepsilon_{\mathrm{a}}\sigma(T_{\mathrm{a}} + 273.15)^4\right] - \varepsilon\sigma(T_{\mathrm{s}} + 273.15)^4, \tag{22}$$

where $r$ is the water reflectivity for longwave radiation, $\varepsilon$ and $\varepsilon_{\mathrm{a}}$ are the emissivities of the lake surface and the atmosphere, respectively; $T_{\mathrm{s}}$ is the water surface temperature (°C), $T_{\mathrm{a}}$ is the air temperature at 2 m height (°C), and

$\sigma = 5.67 \times 10^{-8}\,\mathrm{W\,m^{-2}\,K^{-4}}$ is the Stefan–Boltzmann constant. The reflectivity and emissivity of water are assumed to be 0.04 and 0.96, respectively (e.g., Sun et al., 2007). The emissivity of the atmosphere depends on the water vapor and atmospheric temperature profile. Assuming a standard atmosphere, Brutsaert (1975) derived a formula for calculating the atmospheric emissivity under clear-sky conditions:

$$\varepsilon_{ac} = 1.24 \left[ e_a / (T_a + 273.15) \right]^{1/7}, \tag{23}$$

where $e_a$ is the water vapor pressure (hPa), which is related to the relative humidity (rh) and saturation vapor pressure ($e_s$):

$$e_a = e_s(T_a)\mathrm{rh}. \tag{24}$$

To calculate the saturation water pressure, we use the formula from Winslow et al. (2001):

$$e_s(T_a) = 6.11 \exp \left[ mT_a / (n + T_a) \right]$$
$$\begin{array}{llll}
\text{for} & T_a > 0\,^\circ\mathrm{C} & m = 17.269 & n = 237.7 \\
\text{for} & T_a < 0\,^\circ\mathrm{C} & m = 21.753\ \text{TS7} & n = 265.3.
\end{array} \tag{25}$$

Although other empirical formulas for calculating atmospheric emissivity are available, Brutsaert's (1975) expression (Eq. 23) was reported as the best in many studies of different climates (Wang and Dickinson, 2013). Because Eq. (23) refers to clear-sky conditions, it is necessary to additionally account for cloud effects. Assuming that the emissivity of the water droplets in the clouds is approximately equal to 1, Crawford and Duchon (1999) calculate the total atmospheric emissivity as follows:

$$\varepsilon_a = (1 - f)\varepsilon_{ac} + f, \tag{26}$$

where $f$ is the cloud fraction term defined using the ratio of the previously estimated surface shortwave radiation and surface clear-sky shortwave radiation:

$$f = 1 - S_{surf} / (\tau_{cf} S_{top}). \tag{27}$$

For clear-sky conditions, the cloud fraction term equals 0. However, since the ratio of the surface solar irradiance to the clear-sky irradiance never reaches 0, the cloud fraction term never reaches the theoretical maximum of 1, even in total cloud cover conditions. Note that, even though the model will be run with a time resolution of one hour, the daily mean atmospheric emissivity will be used.

Equation (26) is considered the best formula in many studies (Wang and Dickinson, 2013). By substituting Eqs. (23) and (26) in Eq. (22), we obtain the expression for calculating the net longwave radiation:

$$L_n = \varepsilon \left\{ \left[ (1 - f)\varepsilon_{ac} + f \right] \sigma (T_a + 273.15)^4 \right\}$$
$$- \varepsilon \sigma (T_a + 273.15)^4. \tag{28}$$

### 3.1.3   Latent and sensible heat flux

To calculate the latent and sensible heat flux, we use a slightly modified algorithm provided by Verburg and Antenucci (2010). Their code, which is publicly available at the National Institute of Water and Atmospheric Research (NIWA) website (NIWA, 2021), uses the bulk aerodynamic method based on the Monin–Obukhov similarity theory (Monin and Obukhov, 1954). According to this method, the sensible and latent heat fluxes can be calculated as follows:

$$H_s = -\rho_a c_a C_H U_Z (T_s - T_a), \tag{29}$$
$$H_l = -\rho_a L_v C_E U_Z (q_s - q_a), \tag{30}$$

where $C_H$ and $C_E$ are the transfer coefficients for sensible and latent heat flux, respectively; $c_a = 1005\,\mathrm{J\,kg^{-1}\,K^{-1}}$ is the specific heat of air, $L_v \approx 2500\,\mathrm{kJ\,kg^{-1}}$ is the latent heat of evaporation, $\rho_a$ is the air density ($\mathrm{kg\,m^{-3}}$), and $q_s$ and $q_a$ are the specific humidities ($\mathrm{kg\,kg^{-1}}$) at the water surface and measuring levels, respectively. Air density and specific humidity were determined from the ideal gas law equation and from the observed relative humidity, respectively.

The transfer coefficients were calculated in an iterative procedure, initially assuming neutral atmospheric conditions:

$$C_D = k^2 / \left[ \ln(h/z_0) - \psi_M \right]^2, \tag{31}$$
$$C_E = k^2 / \left\{ \left[ \ln(h/z_0) - \psi_M \right] \left[ \ln(h/z_E) - \psi_E \right] \right\}$$
$$= k C_D^{1/2} / \left[ \ln(h/z_E) - \psi_E \right], \tag{32}$$
$$C_H = C_E, \tag{33}$$

where $C_D$ is the drag coefficient, $h$ is the height above ground (m), $z_0$ and $z_E$ are the roughness lengths (m), and $\psi_M$ and $\psi_E$ are the stability functions for momentum and vapor, respectively. The stability functions are defined through the stability parameter $\zeta = h/L$, where $L$ is the Monin–Obukhov length:

$$L = \frac{-\rho_a u_*^3 T_V}{kg \left( \frac{H_s}{c_a} + 0.61 \frac{(T_a + 273.16\,\text{TS8}) H_l}{L_v} \right)}, \tag{34}$$

where $T_v$ is the virtual temperature. Obviously, $L$ depends on $H_s$ and $H_l$, while $H_s$ and $H_l$ depend on the stability of the atmosphere. Therefore, to calculate $H_s$ and $H_l$, an iterative procedure has to be used. The procedure is initiated by assuming neutral conditions ($\psi_M = \psi_E = 1$). Further details on the calculation of roughness lengths, stability functions, and the iterative process itself can be found in Verburg and Antenucci (2010).

### 3.1.4   Heat brought by precipitation

Assuming the first lake layer in the numerical model gets completely mixed with the precipitation that falls during a time period $\Delta t$ (s), then the temperature of that layer would

equal

$$T_{1+\text{p}} = \frac{\Delta z_1 T_1 + P/(1000 \times 3600)\,\Delta t\,T_{\text{prec}}}{\Delta z_1 + P/(1000 \times 3600)\,\Delta t}, \tag{35}$$

where $T_1$ and $T_{1+\text{p}}$ represent the water temperature of the first layer before and after the precipitation has been introduced (°C), $T_{\text{prec}}$ is the precipitation temperature (°C), $\Delta z_1$ is the thickness of the first layer (m), and $P$ is the hourly precipitation (mm h$^{-1}$). The heat flux brought in by precipitation $H_\text{p}$ (W m$^{-2}$) can then be calculated as

$$\begin{aligned}
H_\text{p} &= \frac{1}{\Delta t}\left[\Delta z_1 + P\,(1000 \times 3600)\,\Delta t\right]\rho c_\text{p}\left(T_{1+\text{p}} - T_1\right) \\
&= \rho c_\text{p} P\,(1000 \times 3600)\left(T_{\text{prec}} - T_1\right).
\end{aligned} \tag{36}$$

Since $T_{\text{prec}}$ was not available, the air temperature was used instead.

## 3.2 Convective mixing

During the night, the net heat flux at a lake surface is generally negative. Consequently, unstable lake stratification is established. However, this unstable stratification is short lived, because the higher density water forming on top of the lake quickly sinks and mixes with the lower density water below it, thus restoring equilibrium (i.e., minimum potential energy).

As Sun et al. (2007) pointed to the importance of introducing a convective mixing mechanism in a water temperature model, we also incorporated this mechanism in the present model. Namely, after each time step of integration, the model algorithm checks whether the upper layer in each pair of two adjacent layers has a higher density than the lower layer. If this occurs, then the two layers are assumed to mix completely, which results in uniform temperature:

$$\begin{aligned}
T_{j\_\text{new}} &= T_{j+1\_\text{new}} \\
&= \left(T_j \Delta z_j + T_{j+1}\Delta z_{j+1}\right)/\left(\Delta z_j + \Delta z_{j+1}\right), \tag{37}
\end{aligned}$$

where $\Delta z_j$ and $\Delta z_{j+1}$ represent the thickness of the $j$th (upper) and $(j+1)$th (lower) layers, respectively; $T_j$ and $T_{j+1}$ are the water temperatures in these layers before convective mixing, respectively; $T_{j\_\text{new}}$ and $T_{j+1\_\text{new}}$ are the temperatures in these layers after convective mixing, respectively.

## 3.3 Model setup

The model code is written in MATLAB programming language. Equation (1) is discretized using the backward Euler scheme:

$$\begin{aligned}
&\frac{c_\text{p}\rho_j}{\Delta t}\left(T_j^{n+1} - T_j^n\right) \\
&= \frac{1}{\Delta z_j}\left[\left(k_\text{m} + k_{\text{t},j+1/2}\right)\left(\frac{T_{j+1}^{n+1} - T_j^{n+1}}{z_{j+1} - z_j}\right)\right. \\
&\quad \left. -\left(k_\text{m} + k_{\text{t},j-1/2}\right)\left(\frac{T_j^{n+1} - T_{j-1}^{n+1}}{z_j - z_{j-1}}\right)\right] \\
&\quad - \frac{1}{\Delta z_j}\left(\phi_{j+1/2}^n - \phi_{j-1/2}^n\right), \tag{38}
\end{aligned}$$

where the subscript denotes the layer or the boundary between two layers, and the superscript denotes the time increment. Notice that the convective term from Eq. (1) is omitted in Eq. (38), CE1 since the algorithm employs convective mixing in a separate procedure after the integration step only if density inversion is detected in the water column, as explained in Sect. 3.2 and shown in Fig. 4. After the stability check, the algorithm performs a step which limits the temperature minimum to 0 °C. Namely, as the model does not include an ice formation module, this step roughly assures no unreasonably low temperatures appear (Fig. 4). Equation (38) can be rearranged as follows: TS9

$$\begin{aligned}
&T_{j-1}^{n+1}\left(\frac{-k_\text{m} - k_{\text{t},j-1/2}}{z_j - z_{j-1}}\right) + T_j^{n+1}\left(\frac{\Delta z_j c_\text{p}\rho_j}{\Delta t}\right. \\
&\quad \left.+ \frac{k_\text{m} + k_{\text{t},j+1/2}}{z_{j+1} - z_j} + \frac{k_\text{m} + k_{\text{t},j-1/2}}{z_j - z_{j-1}}\right) \\
&\quad + T_{j+1}^{n+1}\left(\frac{-k_\text{m} - k_{\text{t},j+1/2}}{z_{j+1} - z_j}\right) \\
&= \left(\frac{\Delta z_j c_\text{p}\rho_j}{\Delta t}\right)T_j^n + \left(\phi_{j+1/2}^n - \phi_{j-1/2}^n\right). \tag{39}
\end{aligned}$$

Equation (39) can be written in matrix form as follows: TS10

$$\mathbf{M}\,T^{n+1} = \mathbf{A}\,T^n + \mathbf{B}. \tag{40}$$

Then, the solution for $T^{n+1}$ is obtained as follows:

$$T^{n+1} = \mathbf{M}^{-1}\mathbf{A}\,T^n + \mathbf{M}^{-1}\mathbf{B}. \tag{41}$$

The implicit Euler scheme is unconditionally stable and thus does not have an upper limit for the time increment. Considering the time resolution of the available input data, the model was run with a time step of one hour (runs with finer time steps were attempted; however, the performance improvements were not significant). The vertical resolution in the model corresponds to the measuring depths and decreases with lake depth. The depths of the integration points were consistent with the sensors' depths, while the boundaries of the layers were set halfway between each pair of consecutive points (Fig. 4b). The layer thicknesses ranged from 0.35 (surface layer) to 16 m (bottom layer). An overview of the model workflow is given in Fig. 4a.

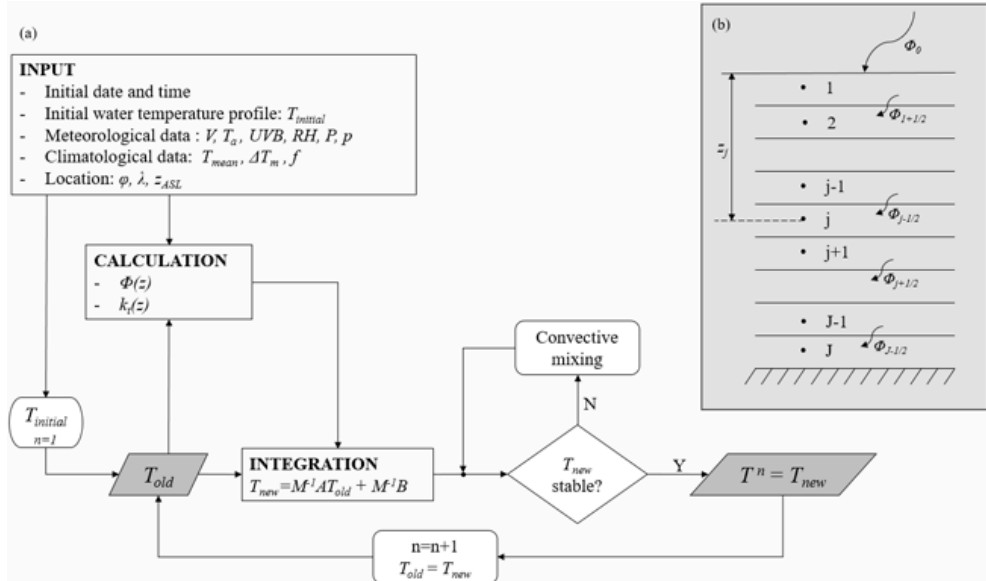

**Figure 4.** Model configuration. Panel **(a)** shows a schematic overview of the model workflow. The input consists of the initial time and date, the initial water temperature profile $T_{\mathrm{initial}}$, meteorological data (wind speed, air temperature, UVB radiation, relative humidity, precipitation, and atmospheric pressure), climatological data (mean annual air temperature, mean annual temperature range between the daily air temperature maximum and minimum, and mean monthly cloud cover), and location data (location latitude, longitude, and altitude above sea level). Panel **(b)** shows the layer setup. Points from 1 to $J$ indicate the integration points where water temperatures are calculated, and $z_j$ is the depth of the $j$th point. The horizontal lines indicate boundaries between layers, $\phi_{j\pm 1/2}$ are the heat fluxes across the layer boundaries, and $\phi_0$ is the net surface heat flux.

## 4   Measures of the model performance

First, a sensitivity analysis was performed to assess the dependence of the model performance on the simulation length. A simulation run was initiated in every hour of the periods with available data, and each was run for up to 30 d. Measured water temperature profiles were used for simulation initialization. Predicted water temperatures and vertical temperature gradients obtained in each simulation after a certain amount of simulation time (from 1 to 30 d) were compared with the corresponding observed values. The results of this analysis are to show the model's ability to provide quality short-term prognoses and the rate of the result deterioration with the increase of the simulation length.

The model performance for each simulation length was evaluated by common bivariate measures. The mean bias error (MBE) is used to assess the tendency of the model to over- or underpredict the temperature. The mean absolute error (MAE) and the root mean square error (RMSE) both provide information about the error central tendency. However, RMSE also accounts for the distribution of the error and becomes larger as the error variability increases. RMSE places more weight on large errors, which makes it more sensitive to outliers. Due to all of the above, it has been argued that MAE is a more natural measure of average error than RMSE (Willmott and Matsuura, 2005). The maximum absolute error (MaxAE) is not a measure of systematic error, but it was

calculated as a measure showing the most extreme outlier. The above measures are calculated from the following expressions:

$$\mathrm{MBE} = \frac{1}{n}\sum_i (P_i - O_i), \tag{42}$$

$$\mathrm{MAE} = \frac{1}{n}\sum_i (|P_i - O_i|), \tag{43}$$

$$\mathrm{RMSE} = \left[\frac{1}{n}\sum_i (P_i - O_i)^2\right]^{1/2}, \tag{44}$$

$$\mathrm{MaxAE} = \max\{|P_i - O_i|\}, \tag{45}$$

where $O$ and $P$ correspond to the observed and predicted values, respectively, while $n$ is the number of corresponding pairs of these values.

The index of agreement values were calculated using three different formulas proposed by Willmott et al. (2012), namely, the original (IA$_{\mathrm{orig}}$), modified (IA$_{\mathrm{mod}}$), and refined

(IA$_{ref}$) index of agreement:

$$IA_{orig} = 1 - \frac{\sum_i [(P_i - \overline{O}) - (O_i - \overline{O})]^2}{\sum_i (|P_i - \overline{O}| + |O_i - \overline{O}|)^2}$$

$$= 1 - \frac{\sum_i (P_i - O_i)^2}{\sum_i (|P_i - \overline{O}| + |O_i - \overline{O}|)^2}, \tag{46}$$

$$IA_{mod} = 1 - \frac{\sum_i |P_i - O_i|}{\sum_i (|P_i - \overline{O}| + |O_i - \overline{O}|)}, \tag{47}$$

$$IA_{ref} = 1 - \frac{\sum_i |P_i - O_i|}{2\sum_i |O_i - \overline{O}|} \text{ for } \sum_i |P_i - O_i|$$

$$< 2\sum_i |O_i - \overline{O}|, \tag{48a}$$

$$IA_{ref} = \frac{2\sum_i |O_i - \overline{O}|}{\sum_i |P_i - O_i|} - 1 \text{ for } \sum_i |P_i - O_i|$$

$$> 2\sum_i |O_i - \overline{O}|. \tag{48b}$$

The IA represents a measure of the relative covariability of the observed and predicted values with respect to the observed mean. The original IA (Eq. 46) uses the square of the difference between predicted and observed values, which is why it overestimates the influence of large errors, similar to the RMSE, which is why the square is replaced with an absolute value in the modified version (Eq. 47); thus, IA$_{mod}$ is less sensitive to outliers than IA$_{orig}$. IA$_{mod}$ approaches 1 (perfect agreement) more slowly than IA$_{orig}$, which means that IA$_{mod}$ is more conservative and allows for finer comparisons of different models with relatively good performance. In IA$_{ref}$ (Eq. 48), the prediction variability in the denominator is replaced with the observation variability. IA$_{orig}$ and IA$_{mod}$ range from 0 to 1, where a value of 0 means that the prediction and observation variabilities are out of phase, while a value of 1 means perfect fit. IA$_{ref}$ ranges from $-1$ to 1 and has a well-defined lower boundary (Eq. 48b), which allows for a better comparison of models with poor performance. However, it should be stressed that IA$_{ref}$ approaching the value of $-1$ does not necessarily indicate poor model performance, because it can also be a result of low observation variability.

The second goal of this study was to examine the ability of the model to predict the springtime onset of lake stratification assuming that there are no measured water temperature data available. For this purpose, a simulation initiated with approximately constant water temperature throughout the entire lake column, which is characteristic of the period when a lake is mixed, was run for the entire year, starting from 1 January 2019. Although accurate results were not expected for the yearlong simulation, the goal of this analysis was to evaluate the extent to which the model can provide relevant information regarding the stratification and/or thermocline depth. Such an approach is particularly appealing for lakes that are completely mixed during the winter, since it does not require measurement of the water temperature profile to initiate the simulation.

## 5 Results and discussion

Based on sporadic observations of the vertical temperature profiles in the Plitvice Lakes, previous studies suggest that Lake 1 and Lake 12 are dimictic (Klaić et al., 2018). Dimictic lakes are covered by ice during winter; they mix in spring and fall; and they are stratified in summer. The continuous observation data of the vertical temperature profiles in Lake 1 and Lake 12, shown in Fig. 2a and c, clearly illustrate for the first time that, during the field campaign, both lakes behaved as warm, monomictic lakes. Specifically, they were mixed during winter but stratified at other times. Furthermore, monomictic lakes (which are frequently found in temperate and tropical latitudes) typically do not freeze, and the two studied lakes did not freeze during the entire field campaign, since the wintertime temperatures in the top lake layers were above 0 °C (Fig. 2b and d).

As the main driver of the lake temperature profile is the surface heat flux, it is interesting to first analyze its terms. Figure 5 shows the modeled mean diurnal variation in the total heat flux and the heat flux terms for a typical winter (a) and summer month (b). The solar heat flux is an order of magnitude higher than the other components of the total heat flux, which indicates that it is one of the main factors affecting the lake water temperature. Next in magnitude is the net longwave radiation, followed by the latent heat flux. The last two components are negative and are responsible for the negative heat flux, or cooling, at night.

The observed and predicted water temperatures for various simulation lengths for 2019 are shown in Fig. 6 (Lake 1 – note that lake temperature measurements started in July) and Fig. 7 (Lake 12). The model performed reasonably well. Namely, the onset of the stratification period (Fig. 7) and both the vertical temperature profile and deepening of the thermocline over time were well captured (Figs. 6 and 7). Simulation results for Lake 12 reproduce the observed data more closely, while for Lake 1, higher discrepancies between simulated and observed data are present, especially for simulation lengths above 10 d. For Lake 1, the position of the maximum temperature gradient in the metalimnion between 12 and 16 m depth was captured, even in the 30 d simulations (Fig. A3), but the temperatures in the epilimnion are significantly overestimated in the stratification period (August) in the longer runs (Figs. 6 and A1).

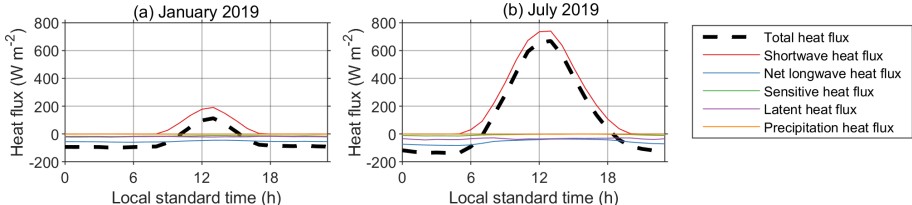

**Figure 5.** Modeled mean diurnal variations in the heat flux at the surface of Lake 12 for January **(a)** and July 2019 **(b)**.

For Lake 12, the difference between the predicted and observed position of the maximum temperature gradient is within 2 m, even for the 30 d simulations, but generally it is lower. Temperature overprediction is noticed in the epilimnion, especially towards the end of the year, for the simulation lengths above 10 d. The stratification began on 21 March, and in the 30 d simulations it was predicted on 23 March; the convective overturn began on 6 September, while in the 30 d simulation it was predicted on 10 September.

Figure 8 shows a closer view of the observed vs. predicted temperatures at depths of 0.2, 5, 15, and 27 m for the period between 6 July 2019 and 31 December 2019. This period was chosen because it is the longest period in which all necessary data (both meteorological and water temperatures for both lakes) were available. Additionally, observed vs. predicted temperature gradients and the prediction errors for both temperature profiles and temperature gradients for the same sample period are presented in Appendix A. As expected, the departure of the predicted from the observed quantities increases with the length of the simulation period. However, even the longest simulation runs (30 d) produced qualitatively acceptable results. Departures of the predicted hourly mean temperatures were mainly $\leq \pm 2$ and $\leq \pm 1\,°C$ for Lake 1 and Lake 12, respectively, except in the thermocline region, where they were mainly $\leq \pm 4$ and $\leq \pm 2\,°C$ for Lake 1 and Lake 12, respectively. The temporal temperature variations at various depths were satisfactorily simulated (Figs. 8, A1, and A2). Furthermore, thermocline depths and their deepening in time were well captured by the model (Figs. A3 and A4). However, the results also suggest that the lake temperatures are somewhat overpredicted in the epilimnion and, at times, may be slightly underpredicted in the hypolimnion (Figs. 8, A1 and A2).

Although the model satisfactorily reproduced the temporal variations in the lake temperatures at the hourly scale, it was not able to reproduce the internal seiches that were previously documented for both lakes (Klaić et al., 2020a, b). This finding is not surprising, since the present model is based solely on the energy budget approach; thus, except for vertical mixing of the two adjacent layers under unstable stratification, it does not account for any hydrodynamic behavior.

Figures 9 and 10 show the calculated model performance measures for both lakes. The model overestimates the water temperature in the epilimnion, especially near the surface and the thermocline region, with an MBE from 0.3 and $<0.1\,°C$ for 1 d simulations and up to 2.6 and 1.2 °C (at 5 m depth) for 30 d simulations in Lake 1 and Lake 12, respectively (Figs. 9a and 10a). The MAE in the epilimnion in Lake 1 starts from $<0.4\,°C$ for 1 d simulations and increases relatively steadily to 2.6 °C for 30 d simulations (Figs. 9b). In Lake 12, it also starts from $<0.4\,°C$ for 1 d simulations and slowly increases to 1.2 °C as the simulation length reaches 30 d (Fig. 10b).

A couple of factors could lead to overestimated temperatures in the upper lake layers. The first is the underestimation of turbulent mixing and turbulent heat transfer, especially in periods of high winds. As seen from Figs. A1 and A2, this overestimation of the uppermost part of the lake is more pronounced for Lake 1 than for Lake 12. As argued in Sect. 2.2.2, measuring site M (where the data used for the atmospheric forcing of the model are measured) is more representative for Lake 1 than for Lake 12. Accordingly, due to its higher altitude and less sheltered position, Lake 1 is more likely exposed to winds stronger than those measured at site M, and thus, both the turbulent mixing and the consequent heat transfer are likely to be stronger than modeled.

The second possible reason is the overestimation of the shortwave radiation extinction coefficient. This coefficient depends on the amount of dissolved organics and particulates in the lake water and can thus be calibrated to reproduce the lake physical properties more closely. We did not proceed with extinction coefficient calibration, because our goal was to investigate the model performance and its general applicability without location-specific fitting.

Also, it is possible that the surface heat flux has been overestimated, as the simplified approach used for its calculation is characterized by limited reliability. Finally, it should be pointed out that the influence of the tributary was not considered, and in case of Plitvice Lakes, it may be non-negligible.

In the hypolimnion, the values of the MBE, MAE, RMSE, and MaxAE remain particularly low, especially for the deepest layers, for both lakes regardless of the simulation length (Figs. 9 and 10). These low values are a result of the low temperature variability in the deep lake layers, which is not taken into account in the formulation of these measures. In Lake 12, the MBE in the hypolimnion stays below 0.5 °C, and in

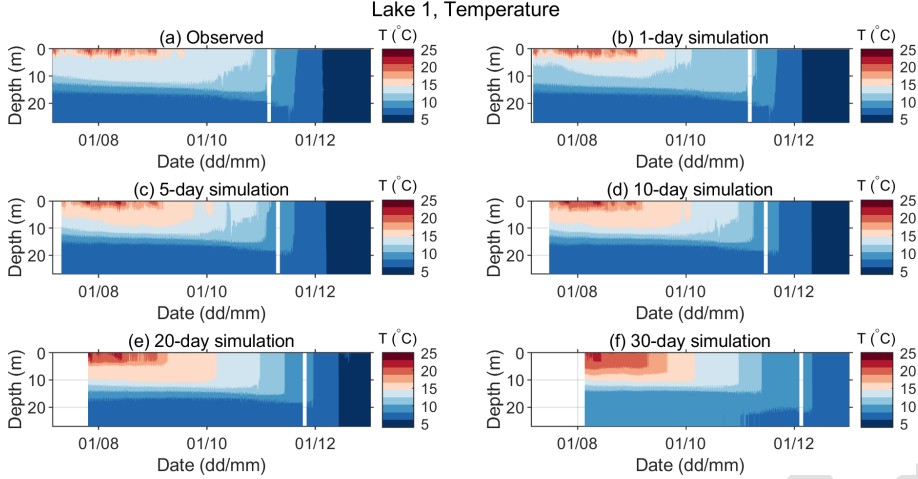

**Figure 6.** Observed **(a)** and predicted **(b–f)** water temperatures of Lake 1 for different simulation lengths in the period between 6 July 2019 and 31 December 2019. Periods with missing data are seen as white vertical stripes.

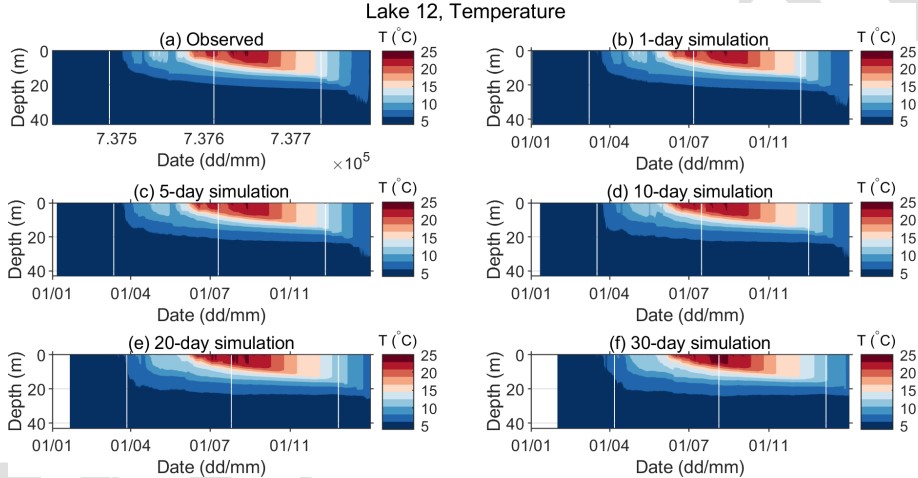

**Figure 7.** Observed **(a)** and predicted **(b–f)** water temperatures of Lake 12 for different simulation lengths for 2019. Periods with missing data are seen as white vertical stripes.

Lake 1, it stays below $1\,°C$. On the other hand, regardless of the formulation (original, modified, or reference), the IA takes the temperature variability into account and therefore decreases with the increase of the simulation length, even in the deep layers, indicating poorer performance as the simulations get longer.

Further inspection of the results for temperature and temperature gradients in Lake 12 (Figs. A2 and A6) shows that the temperature prediction in the metalimnetic layer (thermocline region), where the temperature gradients were the highest, was rather challenging. The model performed relatively poorly in this region, which is particularly noticeable for longer simulation periods.

As seen from Figs. 9 and 10, MaxAE did not increase significantly with increasing simulation length for either of the two lakes. As expected, MaxAE was highest near the surface,

and the maximum values for both lakes were relatively high ($6.7\,°C$ at $3\,m$ depth and $4.9\,°C$ in the surface layer for Lake 1 and Lake 12, respectively).

Figure 11 shows the monthly means of the observed and modeled vertical temperature profiles for Lake 12. The results for Lake 1 are not shown, because the necessary observation data were not available throughout any complete year during the observational campaign. The model successfully reproduced the annual variation in the temperature profile throughout 2019, including the stratification onset and its termination and the thermocline deepening over time. The model underpredicts the surface temperature in January and February (Fig. 11a and b). As the heating starts in spring and continues through summer, the model tends to slightly overpredict the temperatures (Fig. 11e–h). The difference between the predicted and observed surface temperature in the

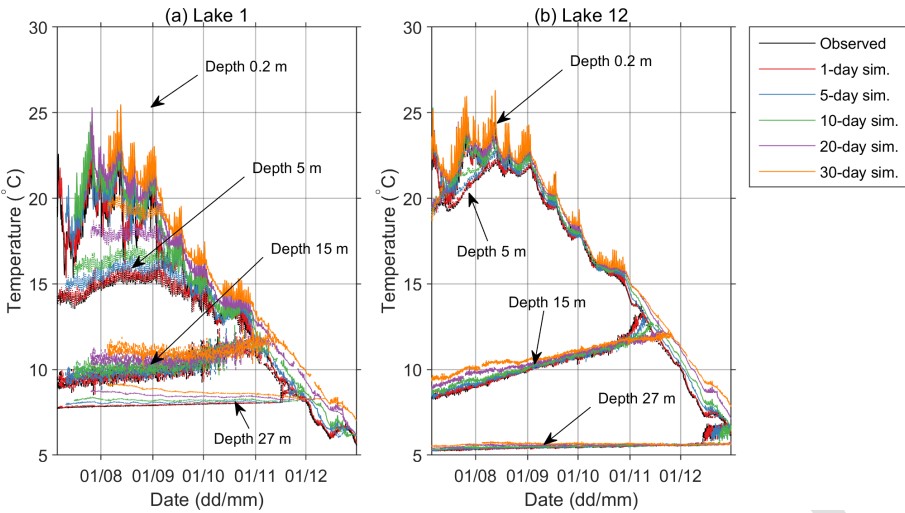

**Figure 8.** Observed and predicted water temperatures at depths of 0.2, 5, 15 and 27 m for different simulation lengths in **(a)** Lake 1 and **(b)** Lake 12 in the period between 6 July 2019 and 31 December 2019.

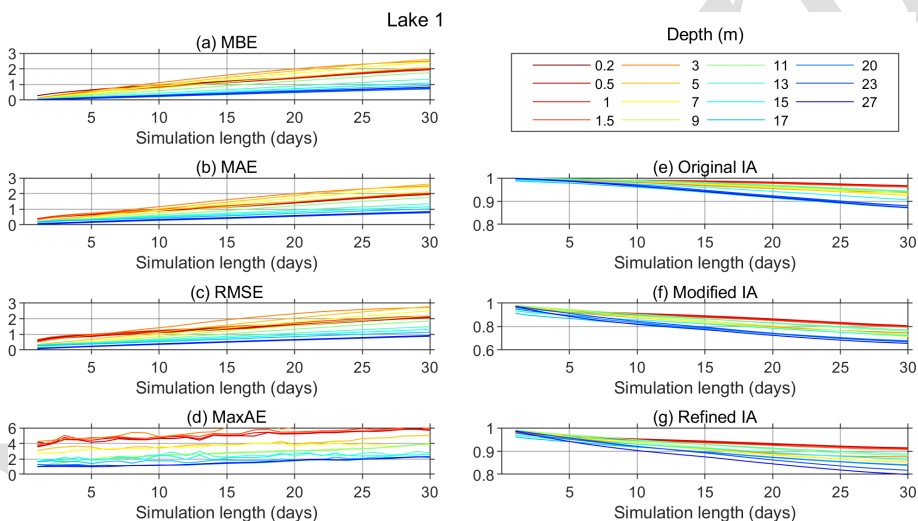

**Figure 9.** Model performance parameters for Lake 1 (calculated for all the periods with necessary data available: 6 July–31 December 2019 and 2 July–30 September 2020).

summer months stays below 1 °C for the longest simulation period. This finding is consistent with the above presented discussion of the model performance measures. In August (Fig. 11h), although the stratification was still strong, the effects of convective mixing during the night started to affect the monthly mean. In the following months (Fig. 11i–l), the mixing depth grew and reached a maximum depth of approximately 20 m in December (Fig. 11l), while the lake stratification was much weaker than that in previous months. In these months of significant convective overturn, higher deviation of the predicted from the observed epilimnion temperature is observed. It becomes more significant with simulation length and reaches approximately 2 °C for the longest simulation period.

The second goal of this study was to examine the ability of the model to predict the onset and termination of stratification and the deepening of the thermocline by yearlong simulation. Because all necessary data for the entire year were only available for 2019, the first day of the yearlong simulation was set to 1 January 2019. For Lake 12, the simulation was initiated with a nearly constant water temperature profile ($\approx 4$ °C) that was observed for 1 January because these data were available, although a constant temperature of $\approx 4$ °C, which was generally observed over the entire water column (which is typical for the wintertime period, when a lake is mixed), can be used instead.

Figure 12 shows the contour diagrams of the observed and predicted water temperatures for Lake 12. Such results for

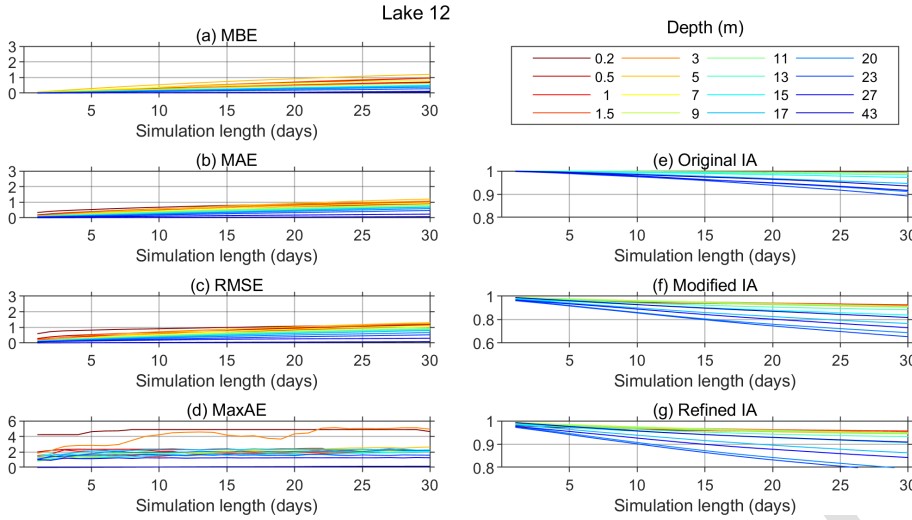

**Figure 10.** Model performance parameters for Lake 12 (calculated for all the periods with necessary data available: 7 July–4 November 2018; 1 January–31 December 2019; 2 July–30 September 2020).

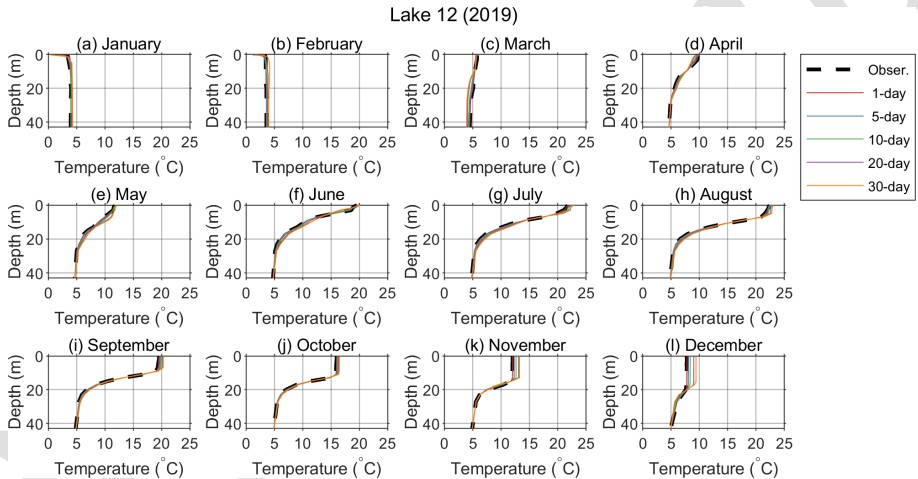

**Figure 11.** Annual variation in the vertical profile of the water temperature. Panels **(a)** to **(l)** show the monthly means of the observed and predicted values in Lake 12 for 2019.

Lake 1 are not shown, because they were almost identical to those obtained for Lake 12. Namely, meteorological forcing drove temperature changes. If the same forcing was used for both lakes, then the only other factor that can introduce a difference in the results was the initial vertical profile, which was very similar for both lakes. As previously pointed out in the discussion of model performance, the model generally overpredicted the temperatures of the upper layers, especially by the end of the year. However, the onset and termination of the stratification period were well predicted, with the onset being captured somewhat better than the termination. The first noticeable temperature rise and the early beginning of stratification appear on 21 March in the observed data and 18 March in the predictions (Fig. 12). Significant temperature gradients exceeding $2\,°C\,m^{-1}$ appear on 12 June; how-

ever, the maximum gradient appears at a depth of around $2.5\,m$ in the predictions and at depth of $5\,m$ (Fig. 13) in the observations. The thermocline depth increases during the summer, and the maximum temperature gradient appears on 21 September at a depth of $12\,m$ and equals $2.5\,°C\,m^{-1}$. On the same date, the maximum predicted temperature gradient appears at the same depth but equals only $1.3\,°C\,m^{-1}$. Actually, Fig. 13 shows that, while the model accurately predicts the upper limit of the metalimnion, it consistently overestimates its thickness, which consequently leads to underprediction of the temperature gradient in it. With temperature gradients below $0.5\,°C\,m^{-1}$, 6 December may be considered as the point of complete end of stratification. Although the predicted mixing depth is in agreement with the measured data, the overestimated epilimnion temperature consequently

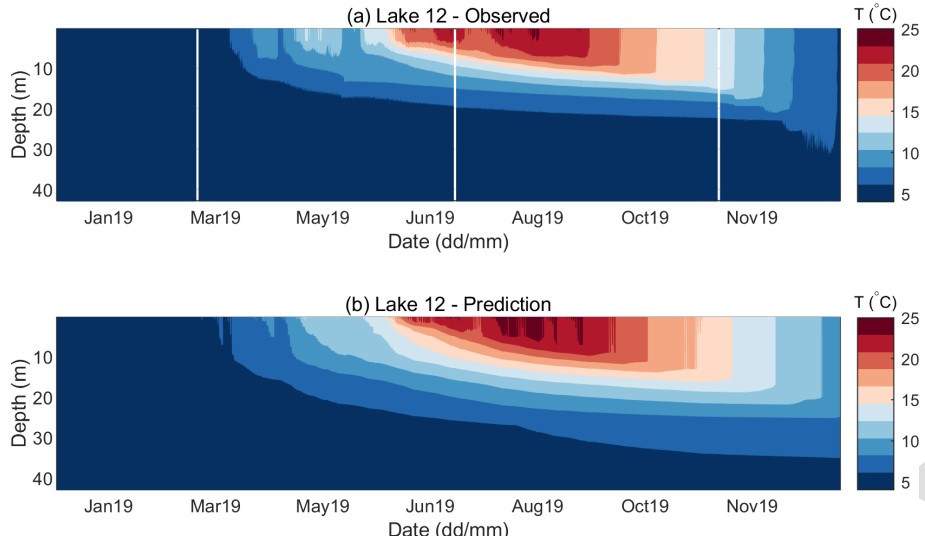

**Figure 12.** Observed **(a)** and predicted **(b)** water temperature for Lake 12 in 2019. The predicted temperature is obtained by a single yearlong simulation run initiated on 1 January 2019.

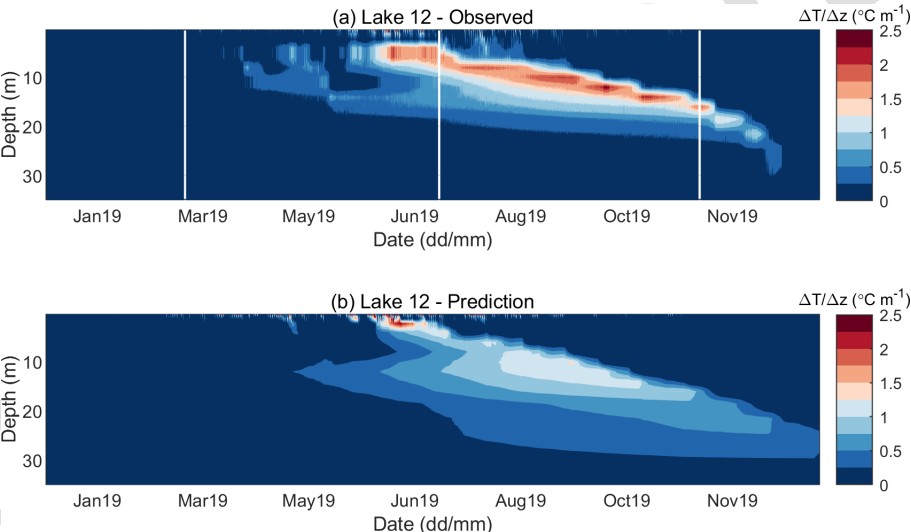

**Figure 13.** Observed **(a)** and predicted **(b)** water temperature gradient for Lake 12 in 2019. The predicted temperature gradient is obtained by a single yearlong simulation run initiated on 1 January 2019.

leads to temperature gradients of around $0.6\,°\mathrm{C\,m^{-1}}$, which persist until the end of the simulation.

This is more clearly presented in Fig. 14, where the observed and modeled monthly profiles are shown. Here, it can also be seen how the model generally overpredicted the monthly mean lake temperatures. The discrepancies between the modeled and observed profiles were largest in the mixed layer during the fall and winter convective overturn. Nevertheless, the mixing depth was well captured. It is concluded that the modeled results satisfactorily reproduced the monthly mean profiles and their annual variation, except af-

ter the convective overturn, when higher temperature overestimation is observed.

## 6 Comparison with other models

To compare the performance of the proposed model with the performance of more complex models, we applied 1-D General Ocean Turbulent Model (GOTM; https://gotm.net/about/, last access: 30 August 2021) version 4.1.0 and Semi-implicit Cross-scale Hydroscience Integrated System Model (SCHISM; Zhang et al., 2016) version 5.3 for Lake 12 for a one-year period, starting from 1 January 2019. GOTM model

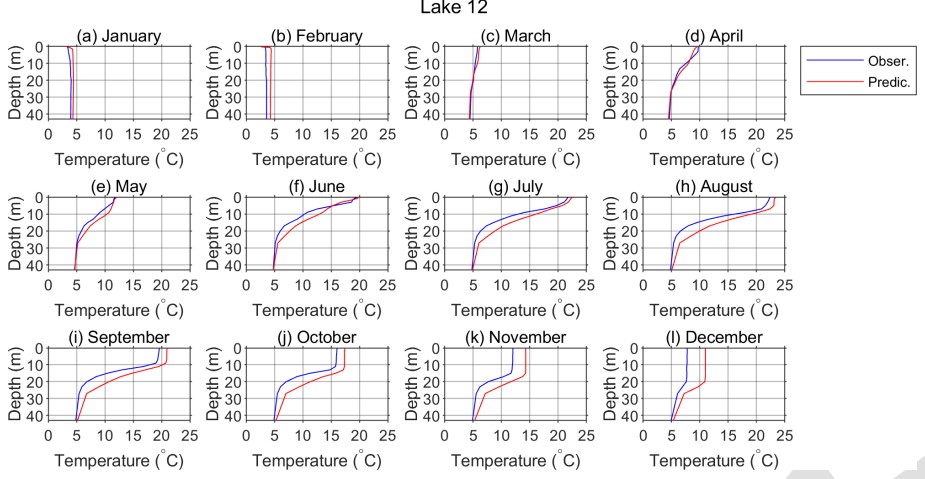

**Figure 14.** Monthly means of the observed (blue) and predicted (red) water temperature vertical profiles for Lake 12 in 2019. Predicted temperatures are obtained by a single yearlong simulation run initiated on 1 January 2019.

is a one-dimensional water column model designed for hydrodynamic, thermodynamic, and biogeochemical studies of lakes and enclosed or semi-enclosed marine water bodies. It simulates vertical transport of momentum, heat, and salt (Burchard et al., 1999). The model, which can be used as a standalone or coupled with other models, has several turbulence closure options. So far, the GOTM model has been applied in a number of oceanographic (e.g., Bruggeman and Bolding, 2014; Burchard et al., 2014; Li et al., 2021) and limnetic studies (e.g., Ciglenečki et al., 2015; Andersen et al., 2021; Nielsen et al., 2021). SCHISM is a three-dimensional (3-D) unstructured-grid model. It employs hydrostatic approximation and solves the Reynolds-averaged momentum as well as the continuity and the transport of salt and heat equations. Due to its unstructured grid, it is suitable for basins with complicated geometry. It has been widely used in hydrodynamic studies of rivers, coastal waters, seas and oceans (e.g., Jacob et al., 2016; Bubalo et al., 2018; Zhang et al., 2020; Burić et al., 2021), and lakes (e.g., Frishfelds et al., 2021). More details on both models and parameterizations employed in the present study are given in Appendix B.

As previously presented, the meteorological forcing for the SIMO simulation was modeled using solely measured data. Apart from the measured air temperature and wind data (GOTM simulation) and measured air temperature (SCHISM simulation), meteorological forcing was modeled with the atmospheric Weather Research and Forecasting (WRF) model (Skamarock and Klemp, 2008). In both GOTM and SCHISM simulations, freshwater was assumed. Also, due to consistency, in both model runs, the same $k$-$\varepsilon$ turbulence closure scheme of Rodi (1984) was employed. Finally, both models were initialized with the lake temperatures observed at 1 January 2019 (same as SIMO).

The comparison of the water temperature at 0.2 m depth, as predicted by the three models (SIMO, GOTM, and

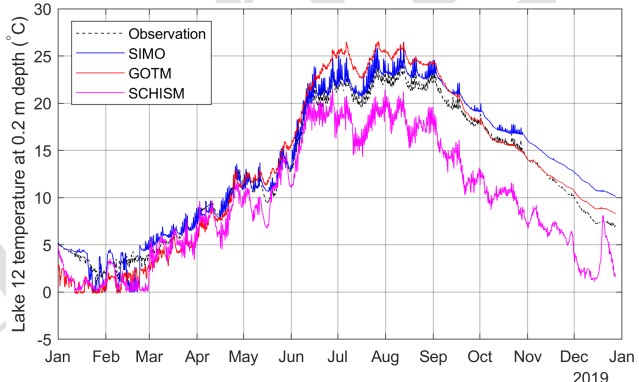

**Figure 15.** Comparison of the near-surface water temperature for SIMO, GOTM, and SCHISM for the period 1 January–27 December 2019.

SCHISM) and the observed values are shown in Fig. 15. SIMO generally outperforms both models until the middle of September, when it starts to consistently overpredict the temperature, while GOTM shows quite low error. On the other hand, SCHISM tends to underestimate the temperature by approximately 5 °C from the beginning of June and almost until the end of the year. The performance measures for the three models are summarized in Table 3, which also shows that SIMO outperforms the other two models considering, all measures except the bias, which is 0.5 °C lower for GOTM.

Numerous lake modeling studies report quantitative performance measures. However, the comparison of model performance with other models is not always straightforward, as there is no common systematic approach. Namely, different studies report different performance measures; sometimes the calculation methods, the observation periods, and the measurement frequency and depths are not clearly stated or measurements are too rare to represent short-term varia-

**Table 3.** Comparison of performance measures for SIMO, GOTM, and SCHISM for the period 1 January–27 December 2019.

| Performance measure | Unit | Model | | |
|---|---|---|---|---|
| | | SIMO | GOTM | SCHISM |
| RMSE | °C | 1.48 | 1.85 | 3.91 |
| Bias | °C | 0.85 | 0.35 | −3.34 |
| MAE | °C | 1.18 | 1.53 | 3.37 |
| MaxAE | °C | 3.96 | 4.41 | 8.24 |
| Original IA | – | 0.99 | 0.99 | 0.92 |
| Modified IA | – | 0.90 | 0.88 | 0.72 |
| Refined IA | – | 0.90 | 0.87 | 0.72 |

tions. Furthermore, no studies calculating the performance measures in relation to the simulation period using only the end results, as done here, were found. However, quite a few studies report on single, longer simulations. Some of these results are summarized in Table 4. For a yearlong simulation of the water temperature in a small dimictic lake, Martynov et al. (2010) reported a surface temperature RMSE of 1.8 °C for an eddy diffusivity model (Hostetler model) and 3.2 °C for a two-layer model (FLake). Bruce et al. (2018) ran a two-year simulation for 32 different lakes using the GLM model, and the calculated RMSEs for the entire vertical profile, epilimnion, and hypolimnion were 1.34, 1.62, and 1.31 °C, respectively. MacKay (2012) ran a bulk mixed model simulation for approximately a month and a half and reported a surface temperature MBE <1 °C. Read et al. (2014) ran a 30-year simulation (restarted annually) for 434 temperate lakes and reported a RMSE of 2.78, 1.74, and 3.33 °C for the entire vertical profile, epilimnion, and hypolimnion, respectively. Moore et al. (2021) ran four different models for a temperate monomictic lake and reported RMSE values from 0.8 to 2.96 °C for the runs before the model parameter calibration and 0.61 to 1.17 °C after it. The reported absolute MBE values ranged from 0.34 to 1.75 °C for the runs before the model parameter calibration and 0.1 to 0.55 °C after it.

The yearlong simulation in this study resulted in a surface temperature RMSE of 1.51 °C; in the hypolimnion, the RMSE was lowest in the deepest layer at 0.33 °C. The RMSE was the highest in the thermocline region, where it reached a maximum of 2.8 °C at 17 m depth. The RMSE for the entire profile was 1.91 °C. The surface temperature MBE was 0.88 °C. The maximum MBE was again in the thermocline region and equaled 2.28 °C. This systematic overprediction can also be noticed in Fig. 14. Considering the lake surface temperature and entire vertical profile as well as the epilimnion and hypolimnion temperature CE2, the model performance for the yearlong simulation was satisfactory, since it was comparable with the performances of other models (Table 4). The performance for the thermocline region was somewhat poorer, but performance in that region was not specifically reported in the reviewed literature.

## 7 Summary and conclusions

The aim of this study was to offer a simple 1-D energy budget model for the prediction of the vertical temperature profile in a small, warm, monomictic lake that is forced by a reduced number of input meteorological variables. Specifically, these include meteorological variables that are routinely measured at meteorological stations (i.e., the air temperature, relative humidity, atmospheric pressure, wind speed, and precipitation) as well as UVB radiation data and climatological yearly mean temperature data. In addition, an observed vertical profile of the lake temperature was used as an initial condition.

The main challenge was to calculate the net heat flux on the lake surface and to determine its components (i.e., the shortwave and longwave radiation, sensible and latent heat flux, and precipitation heat flux) from the available data. The model performance was evaluated using lake temperatures measured continuously during an observational campaign in two lakes of Plitvice Lakes, Croatia: Lake 1 (Prošće Lake) and Lake 12 (Kozjak Lake). The necessary meteorological data were provided by a single meteorological station located approximately 2 and 1.6 km from the lake temperature measuring points for Lake 1 and Lake 12, respectively. Except being further away from the meteorological station, Lake 1 has an altitude approximately 100 m higher than Lake 12, is surrounded by more complex orography, and is very likely exposed to stronger winds and lower air temperatures than those used as meteorological input data. Accordingly, the model performance was somewhat poorer for Lake 1, which indicates the importance of the microlocation-specific input meteorological data, as the meteorological forcing is the main driver of the temperature profile evolution. In addition, the influence of the tributary water that inflows into Lake 1, which was not taken into account in the present model, could also contribute to higher differences between the modeled and measured temperatures in comparison to Lake 12.

Generally, epilimnion temperature was somewhat overestimated, especially with the onset of the convective overturn. The upper limit of the metalimnion was well captured, while its thickness was overestimated, leading to underestimation in the maximal temperature gradient. However, the

**Table 4.** Comparison of SIMO performance with other models.

| Reference | Model | Application area | Simulation length | RMSE | MBE |
|---|---|---|---|---|---|
| | SIMO | Small, monomictic lake | 1 year | 1.91 °C (entire vertical profile) 1.51 °C (surface temp.) 1.95 °C (epilimnion temp.) 1.13 °C (hypolimnion temp.) | 0.88 °C (surface temp.) 1.37 °C (entire vertical profile) |
| | SIMO | Small, monomictic lake | 1.5 months | | 0.33 °C (surface temp.) |
| Martynov et al. (2010) | (a) Hostetler model (b) FLake | Small, dimictic lake | 1 year | (a) 1.8 °C (surface temp.) (b) 3.2 °C (surface temp.) | |
| Bruce et al. (2018) | GLM | 32 different lakes | | 1.34 °C (entire vertical profile) 1.62 °C (epilimnion temp.) 1.31 °C (hypolimnion temp.) | |
| MacKay (2012) | Bulk mixed model | Arctic lake | 1.5 months | | <1 °C (surface temp.) |
| Read et al. (2014) | GLM | 434 temperate lakes | 30 years (restarted annually) | 2.78 °C (entire vertical profile) 1.74 °C (epilimnion temp.) 3.33 °C (hypolimnion temp.) | |
| Moore et al. (2021) | (a) FLake (b) GLM (c) GOTM (d) Simstrat | Temperate, monomictic lake | 1 year | (a) 2.96/0.61 (b) 0.94/1.17 (c) 0.80/0.85 (d) 1.10/0.70 (not calibrated/calibrated) (entire vertical profile) | (a) −1.75/−0.3 (b) −0.34/0.10 (c) −0.49/−0.55 (d) 0.57/−0.35 (not calibrated/ calibrated) (entire vertical profile) |

model satisfactorily estimated the stratification and overturn dynamics. There are several possible causes of departures of modeled from measured temperatures. One of them is the underestimation of the turbulent heat transfer in the epilimnion, especially in periods of high winds. In addition, the model cannot simulate internal seiches and possible water exchange between the warmer epilimnion and colder hypolimnion. Other probable causes are the use of an inappropriate light extinction coefficient value and the limited reliability of the surface heat flux. However, considering all model simplifications, we conclude that the model performed reasonably well.

The sensitivity analysis of the model performance to the simulation length showed that, when using appropriate meteorological forcing (as is the case of Lake 12), the model performance, especially in the epilimnion, steadily deteriorated up to a simulation length of approximately 15 d; however, a further increase in the simulation length up to 30 d had little effect on the model performance parameters. This proves the model can be used for obtaining reasonable lake water temperature prognosis for at least 30 d-long periods.

Despite the model's shortcomings, the yearlong simulation showed that the model is able to predict the onset of stratification and convective overturn relatively precisely, and the values of the model performance measures were comparable to those reported for other models. Thus, for a certain lake with no water temperature measurement data available, a yearlong simulation such as this would provide an assessment of lake stratification establishment, which can be useful for various studies dealing with lake biology, geochemistry, sedimentology, etc.

To further corroborate the general applicability of the present model, it should be applied to a larger number of different monomictic lakes. Nevertheless, in the present study, no lake-specific parameter tuning was performed. Thus, we expect similar model performance for other lakes if adequate meteorological forcing is employed.

**Appendix A**

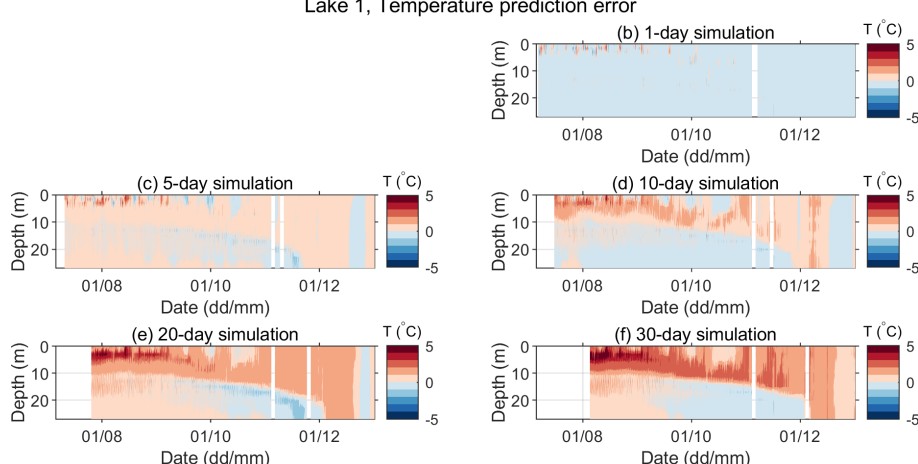

**Figure A1.** Error in the predicted water temperature ($P_i - O_i$) for Lake 1 for different simulation lengths for the period between 6 July and 31 December 2019. Panel **(a)** is omitted so that the panels' positions for different simulation lengths correspond to those in Fig. 6.

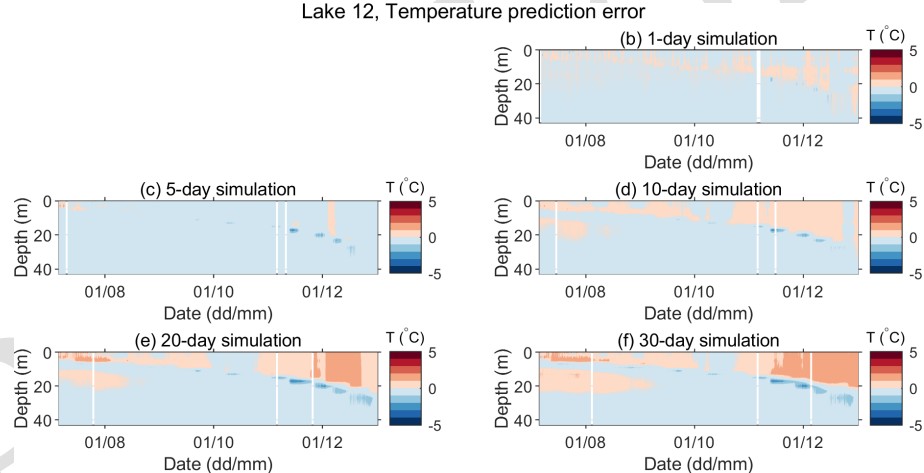

**Figure A2.** Error in the predicted water temperature ($P_i - O_i$) for Lake 12 for different simulation lengths in the period between 6 July and 31 December 2019. Panel **(a)** is omitted so that the panels' positions for different simulation lengths correspond to those in Fig. 7.

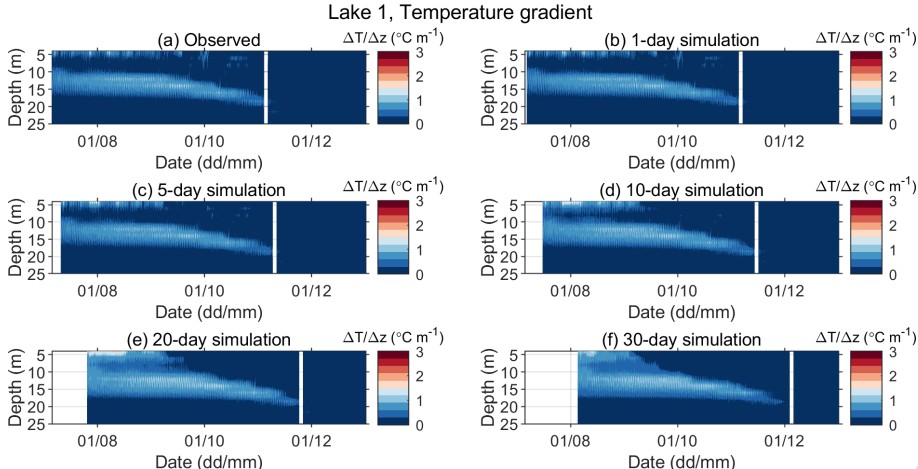

**Figure A3.** Observed **(a)** and predicted **(b–f)** vertical gradients of water temperature for Lake 1 for different simulation lengths in the period between 6 July and 31 December 2019.

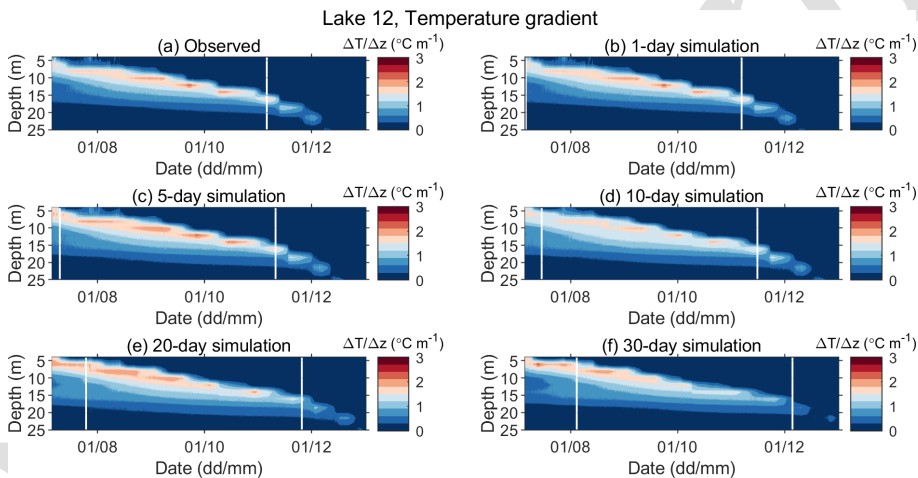

**Figure A4.** Observed **(a)** and predicted **(b–f)** vertical gradients of water temperature in Lake 12 for different simulation lengths in the period between 6 July and 31 December 2019.

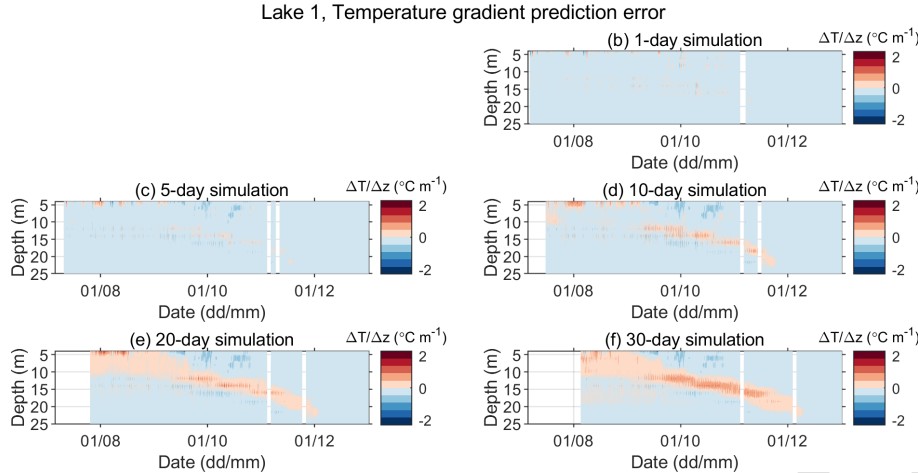

**Figure A5.** Error in the predicted vertical gradient of water temperature ($P_i - O_i$) in Lake 1 for different simulation lengths in the period between 6 July and 31 December 2019. Panel **(a)** is omitted so that the panels' positions for different simulation lengths correspond to those in Fig. 3.

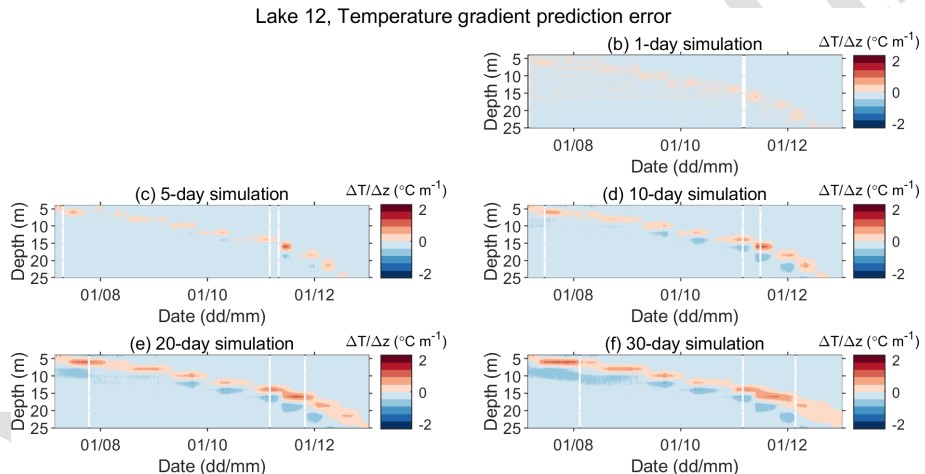

**Figure A6.** Error in the predicted vertical gradient of water temperature ($P_i - O_i$) in Lake 12 for different simulation lengths in the period between 6 July and 31 December 2019. Panel **(a)** is omitted so that the panels' positions for different simulation lengths correspond to those in Fig. 5.

## Appendix B

### B1 Description of the SCHISM and GOTM model parametrization

The hydrodynamic model system SCHISM (Semi-implicit Cross-scale Hydroscience Integrated System Model; Zhang et al., 2016) was designed for simulations of the 3-D baroclinic and barotropic circulation at different spatio-temporal scales. The calculation is conducted on the points of a horizontal unstructured triangular grid, which is one of the most important features of the model, because the use of such a grid ensures a high spatial resolution. In the calculations, the model uses an efficient and accurate semi-implicit method on finite elements or volumes with the Euler–Lagrange algorithm to solve the Reynolds-averaged Navier–Stokes equations (in hydrostatic and non-hydrostatic form) in order to more realistically described a wide spectrum of physical and biological processes as well as atmospheric and river forcing. The equations are simplified by considering the hydrostatic and Boussinesq approximations.

The horizontal grid covers the entire area of 16 cascade lakes, and it is composed of 17 472 elements whose surfaces range from 1 to 200 m$^2$. In the vertical, a hybrid SZ grid was used, whereby the hybrid $Z$ layers are fixed at a certain depth, located below the S coordinates that follow the terrain (Song and Haidvogel, 1994) according to the prescribed distance. The Plitvice Lakes are shallow enough that it is not necessary to define $Z$ layers, and 30 sigma TS11 levels were used in the vertical discretization. During model calibration, i.e., when adjusting various parameters, it turned out that the model gives the best results in simulations with a time step of 10 s. Bottom friction in the model is approximated by the quadratic law of friction, defined by the assigned coefficient of friction with the adopted standard value of 0.003. As the Plitvice Lakes are extremely transparent and clean, in order to simulate a realistic lake character, Jerlov I was taken as the type of water, whose extinction coefficients correspond to the clear water. For the lake albedo, the theoretical values of 6 %, which are usual for the ocean, were applied. CE3 Vertical mixing in the model is imposed through the je CE4 $k$-$\varepsilon$ scheme with the Kantha–Clayson stability function. TVD (total variation diminishing) scheme was used in the advective terms of the transport equation. TVD is a slower scheme but better displays sharp temperature gradients. A baroclinic mode was also included, by which the contribution of temperature to the density of the medium is included in the equations of motion.

GOTM is a 1-D water column model for simulating the most important hydrodynamic and thermodynamic processes related to vertical mixing in natural waters. The GOTM model is suitable for simulating and predicting the stratification and vertical temperature profile of closed systems, such as the Plitvice Lakes. It is configured in such a way that it can be connected to 3-D circulation models, such as SCHISM, and used as a module to calculate vertical turbulent mixing. The core of the model calculates solutions for one-dimensional versions of the momentum, salt, and heat transport equations. The strength of GOTM is in the large number of tested turbulence models implemented in the code. Calculations are made at only one point along the entire vertical where any number of layers can be placed. Its advantage is in its faster performance and better formulation of the heat flow between the atmosphere and water.

Model parameters such as water type and turbulence scheme in GOTM are the same as in the SCHISM model. Jerlov I (clear water) was taken as the type of water, and the $k$-$\varepsilon$ TS12 scheme was used as the turbulent mixing scheme. The number of vertical layers at point K1 (Lake 12) was set to 91, because it is at a greater depth, while 60 layers were taken for point P1 (Lake 1).

Apart from the measured air temperature and wind data (GOTM simulation) and measured air temperature (SCHISM simulation), both models use time series of atmospheric variables from the WRF model and heat fluxes on the surface of the lake, which are the main driver of the physical processes that cause thermal stratification and vertical mixing in the lake. The models are forced by atmospheric input on an hourly basis, with SCHISM additionally having an hourly loop that simulates the exchange of heat, mass, and momentum between the lake and the atmosphere and the consequent heating and mixing processes that occur in the lake.

*Code and data availability.* The SIMO v1.0 code is published under Creative Commons Attribution 4.0 International license and it is available at https://doi.org/10.5281/zenodo.6367810 (Šarović and Klaić, 2021).

Lake water temperature data are available on request for research purposes by contacting Zvjezdana B. Klaić (zklaic@gfz.hr). Authors are not authorized to publicly share meteorological data. To access these data, requests should be sent to the Croatian Meteorological and Hydrological Service. TS13

*Author contributions.* KŠ designed the SIMO model, wrote the model code, ran the simulations, and performed the result post processing. ZBK organized the lake temperature experiment and contributed to the model development and the result evaluation and interpretation. Both KŠ and ZBK contributed to the discussion and writing of the paper. MB produced composite pictures of the lake bathymetries and the digital orthophoto images, ran the GOTM and SCHISM model simulations, performed corresponding post processing, and contributed to writing of Sects. 6 and 7.

*Competing interests.* The contact author has declared that none of the authors has any competing interests.

*Acknowledgements.* This study was performed under the project "Hydrodynamic Modeling of Plitvice Lakes System" founded by the Plitvice Lakes National Park, Croatia. We also appreciate the technical support of the PLNP during equipment installation and lake temperature data acquisition. Meteorological data were provided by the Croatian Meteorological and Hydrological Service. A 2019 bathymetry experiment was organized by the PLNP. We are grateful to Goran Gašparac of Croatia Control Ltd., Velika Gorica for performing WRF model simulations. Finally, we gratefully acknowledge the useful comments of two anonymous reviewers.

*Financial support.* This study was supported by the project "Hydrodynamic Modeling of Plitvice Lakes System" (PLNP; grant no. 7989/16). TS14

*Review statement.* This paper was edited by Bethanna Jackson and reviewed by two anonymous referees.

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

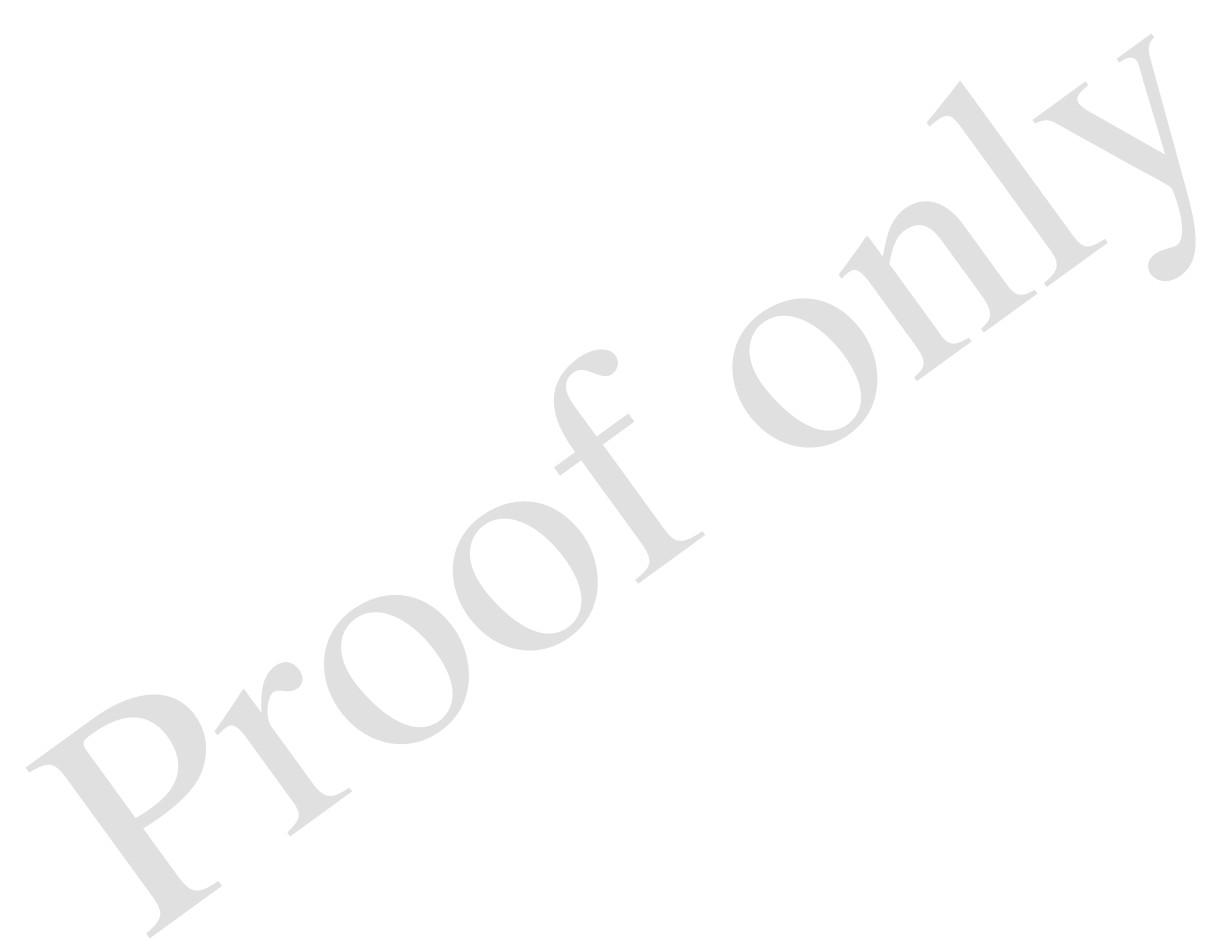

**Remarks from the language copy-editor**

CE1     There is a comment here about a comma, but we are not sure what it is referring to. Please check.

CE2     Adjustment made: please check.

CE3     Adjustment made: please check.

CE4     Please double-check "je" here.

**Remarks from the typesetter**

TS1     Please confirm $\phi$ throughout.

TS2     Please confirm *Pr* throughout.

TS3     Thank you for your feedback. Please note: variables that consist of 2 or more letters have to be roman according to our standards (except special numbers such as "*Ri*).

TS4     Please give an explanation of why the equation needs to be changed. We have to ask the handling editor for approval. Thanks.

TS5     Please confirm equation.

TS6     Please see my previous comment concerning variables that consist of 2 or more letters.

TS7     Please give an explanation of why this number needs to be changed. We have to ask the handling editor for approval. Thanks.

TS8     Please give an explanation of why this number needs to be changed. We have to ask the handling editor for approval. Thanks.

TS9     Please confirm equation.

TS10     Please confirm vectors and matrices.

TS11     Please note: no change has been inserted since there should not be a space in $30\sigma$ according to our standards.

TS12     Please confirm.

TS13     Please confirm this section.

TS14     Please confirm acknowledgements and financial support sections.

TS15     Please note: "C12" has not been added since the issue number is not needed.

TS16     Please confirm reference list entry.