# Peer review of "SIMO v1.0: Simplified model of the vertical temperature profile in a small warm monomictic lake"

_Geoscientific Model Development, 2021_

## Author Comment (AC1)

At the beginning, we would like to thank Referee #1 for useful comments which helped us improve the manuscript. Our response is given bellow.
For easier referring to the each comment we also include the full Referee #1 review in black, together with our answers in blue.

RC1: 'Comment on gmd-2021-118', Anonymous Referee #1, 08 Sep 2021   reply
**General comments**

This manuscript describes a new, comparably simple model for the 1D-simulation of the vertical temperature profiles in a lake. The paper is clearly structured and easy to read, and I appreciate the detailed and self-critical analysis of the model results. However, I have some general concerns about the scientific novelty and approach as well as the scientific rigor of the work.

First, the scientific focus and novelty of the paper remains unclear. It includes two different topics that are, from the scientific point of view, largely independent. And for both of these topics, a different approach would have been more appropriate for a thorough scientific investigation.

The first topic deals with estimating the heat fluxes at the lake surface in the absence of direct measurements of shortwave and longwave radiation. Such parameterizations of the surface heat fluxes have been previously described in numerous publications. If there is any novelty in the approach that the authors use here for this purpose, it is not made clear to the reader. The only thing that I haven't seen in the context of lake modelling is the suggestion to derive the daily dynamics of solar radiation from UV-B measurements. However, this seems to be mainly a workaround for this specific case than a generally applicable approach, as in general, observations of global radiation are much more frequently available than observations of UV-B radiation. Furthermore, the approach used in the study does not allow to check whether the applied heat flux parameterization works well. In fact, the results seem to indicate that it doesn't, given that simulated lake surface temperatures significantly and consistently overestimate observations even in very short model runs of a few days.

Thank you for your comment. We certainly agree that the employed parameterizations of longwave and shortwave radiation have been published previously. However, to our best knowledge, there is no paper addressing the application of these parameterizations to lake modeling, and, this is one of novelties of the present study. To emphasize these more clearly, we added the text *"Although the proposed model employs well known parameterizations of longwave and shortwave radiation, in the present study these parameterizations are built into a lake temperature model for the first time."* in the Introduction.
We agree that in general, global radiation data are more available than UVB data. Nevertheless, global radiation is not routinely observed, and a lack of appropriate data and/or the need for computation of these data is already pointed to by others (e.g., Bell et al., 2006; Martynov et al., 2010; Hipsey et al., 2019). In addition, in the recent decades the interest for monitoring of UVB radiation has been increasing, particularly in touristic areas, due to the awareness of the harmful effects of the ultraviolet radiation (e.g., Kudish et al., 2005; Podstawczyńska, 2010). So, apart from the present case, there might exist places elsewhere where UVB data are available, while global radiation data are not. Furthermore, proposed approach can also be applied for periods for which global radiation data are temporarily unavailable due to equipment malfunction, while UVB data exist.

Based on Reviewers' Specific comment regarding Equation 22 we assumed a constant value for the light extinction coefficient ($\lambda e$ = 0.1 as in Henderson-Sellers, 1986). This resulted in significant improvement of the model results as shown in more details in the revised manuscript.

Nevertheless, to better assess proposed model performance, in the revised manuscript we added a new section "6 Comparison with other models", where we compare SIMO model results with the results of two other, more complex models (GOTM and SCHISM). It is shown that departures of modeled values from observations at the lake surface are comparable for all three models and that SIMO even outperforms the other models in some periods, especially SCHISM. The results are also shown in the table and figure below.

[Figure]

*Figure 1: Comparison of performance measures for SIMO, GOTM and SCHISM for the period from 01.01.-27.12.2019.*

**Table 1 Comparison of performance measures for SIMO, GOTM and SCHISM for the period from 01.01.-27.12.2019.**

| Performance measure | Unit | Model | | |
|---|---|---|---|---|
| | | SIMO | GOTM | SCHISM |
| **RMSE** | ° C | 1,48 | 1,85 | 3,91 |
| **Bias** | ° C | 0,85 | 0,35 | -3,34 |
| **MAE** | ° C | 1,18 | 1,53 | 3,37 |
| **MaxAE** | ° C | 3,96 | 4,41 | 8,24 |
| **Original IA** | - | 0,99 | 0,99 | 0,92 |
| **Modified IA** | - | 0,90 | 0,88 | 0,72 |
| **Refined IA** | - | 0,90 | 0,87 | 0,72 |

The second topic is the temperature model for the given lakes. Again, it is not clearly pointed out what is new about the modelling approach. The model seems to be mostly taken from the paper of Sun et al. (2007) with the addition of a turbulent term. And also here, the approach of the study doesn't really allow to assess how well the model works. This would probably be done best by comparing the simulations with those of other models forced with the same surface heat fluxes, which might then allow to assess to what extent the relatively large discrepancies between simulations and observations are caused by the surface heat flux parameterization and by the actual lake model.

The following text has been added in the introduction: *"Although the proposed model employs well known parameterizations of longwave and shortwave radiation, in the present study these parameterizations are for the first time built into a lake temperature model. Furthermore, in comparison with the model of Sun et al. (2007), the present model does not neglect the turbulent diffusion for small lakes."*

To assess how well proposed model simulates lake temperatures, we compared lake temperatures simulated by SIMO model with those simulated by two more complex models as described in our response above and in the new section "6 Comparison with other models".

Second, the model description and the equations contain several errors that are described in the following detailed comments. I did not check all equations in detail, but some things are clearly wrong. Some of the errors are probably only typos in the text or errors when creating the figures, but others might also be wrong in the model formulation.

Thank you for drawing our attention to typos and errors (luckily, only in equations, while in the model code these were correct). In the revised version these are corrected as stated in our responses to Specific comments.
In addition, we corrected former Eq. 18 by changing $(2\omega_s)^2$ to $(2\omega_s^2)$. We also added the albedo term in the light extinction equation (Eq. 20, former Eq.21), which was omitted in the submitted version of the manuscript (but was correctly used in the code).

For these reasons, I cannot recommend to accept publishing this paper in Geoscientific Model Development in its present form.

**Specific comments**

[1] Line 15: I don't think that it is clear for the reader here what is meant with "a sensitivity analysis of the simulation length"

The sentence has been reformulated to: *"The model performance was evaluated for simulation lengths from 1 to 30 days."*

The explanation in chapter 4 was also expanded: *First, a sensitivity analysis was performed to assess the dependence of the model performance on the simulation length. A simulation run was initiated in every hour of the periods with available data and each was run for up to 30 days. Predicted water temperatures and vertical temperature gradients obtained in each simulation after certain amount of simulation time (from 1 to 30 days) were compared with the corresponding observed values.*

We hope that makes it clearer.

[2] Study area: it would be easier for the reader to have the lake properties in a table rather than in the text.

Thank you for your suggestion. A table containing the lake properties has been created.

[3] Figure 3: is there any specific reason for using J/m2/h rather than the standard W/m2 for UV radiation?

The unit has now been switched to W m$^{-2}$ as suggested.

[4] Line 149: check the usage of phi, there is capital phi in the text and small phi in the equation. Small phi is also used for latitude and capital phi later for the surface heat flux. Please use consistent and unique symbols.

Thank you for noticing. The usage of $\Phi$ has been checked and corrected. In addition, the error in the interpretation of $\Phi$ in former line 149 is corrected. Now it is stated the $\Phi$ represents the heath flux instead of the heat source.

[5] Equation (2): I don't know the source of that equation, as Sun et al (2007) don't give a reference for it, but for high temperatures, the density calculated with this equation seems to be quite far from other standard equations that are usually applied in lake and ocean models (e.g., Chen and Millero or IES-80).

Thank you for your suggestion. The figure bellow shows the water density with respect to the its temperature calculated according to the equation from Sun et al. (2007) and from Chen and Millero (1986) assuming the salinity equals 0. Although we don't expect significant effect on the results we accept the suggestion and we adopted the Chen and Millero (1986) equation.

[Figure]

[6] Equation (3): I think there is a factor z missing in the equation.

Thank you for noticing. The equation has been corrected. Please, note that the mistake was present only in the text and not in the code.

[7] Line 169: I don't think it makes sense to neglect turbulent transport even in lakes shallower than 10 m. This is usually one of the main drivers determining the surface mixed layer depth (e.g. Monismith and MacIntyre, 2009, https://doi.org/10.1016/B978-012370626-3.00078-8 ).

Thank you for your suggestion. The text has been modified accordingly. The paragraph referring to the suggestion of Sun at el. (2007), that for shallow lakes turbulent transport can be neglected, has been removed. The claim was perhaps not entirely correct, as it is not the depth that is the crucial factor but the lake area. Even Monismith and MacIntyre (2009) come to a conclusion that large lakes have deeper mixed layers and that for small temperate lakes the deepening of the mixed layer is dominated by heat loss (a convenient example is shown in Figure 11). An exception are very shallow polymictic lakes where strong stratification doesn't develop.
Nevertheless, please note that turbulent thermal diffusion was already taken into account in the code.

[8] Chapter 3.1.1: It is not clear from the text how exactly the chosen approach accounts for the effect of cloudiness on surface downward solar radiation.

Cloudiness is indirectly taken into account. Winslow et al. (2001) found there is an inverse, almost linear relationship between humidity and daily transmittance. In former eq.(11) $S_{top}\tau_{cs}$ represents the surface solar radiation in clear sky conditions and the factor $D(1-\beta e_s(T_{min})/e_s(T_{max}))$ (in the new version we switched $e_s(T_{min})/e_s(T_{max})$ to $rh_{Tmax}$ as we realized it is more convenient) accounts for the cloudiness.
The text has been changed as follows:

*According to Winslow et al. (2001), the daily solar irradiance at the Earth's surface is equal to*

$$S_{surf} = \tau_{cf} D\left(1 - \beta_s rh_{T_{max}}\right)S_{top} ,$$

(1)

*where $S_{surf}$ is the total daily solar irradiance at the surface (J m$^{-2}$ day$^{-1}$), $\tau_{cf}$ is the cloud-free atmospheric transmittance, $\beta_s$ is an additional parameter required to introduce variation between sites, $rh_{Tmax}$ is the relative humidity at the daily maximum air temperature ($T_{max}$), and $S_{top}$ is the total daily solar irradiance at the top of the atmosphere (J m$^{-2}$ day$^{-1}$).*

*...*

*From Eq. (1), $\tau_{cf} S_{top}$ is the maximum cloud free value of $S_{surf}$. The effect of cloudiness is indirectly taken into account by introducing the factor $(1-\beta_s rh_{Tmax})$, based on the finding that the solar irradiation from sunrise until the maximum daily air temperature is reached ($S_{surf\_Tmax}$), is proportional to the relative humidity at that moment. The factor $D=S_{surf}/S_{surf\_Tmax}$ is introduced to account for the surface solar irradiation from when the air temperature reaches its daily maximum until sunset. D is calculated assuming that the air temperature reaches its maximum around 3pm:*

$$D = \left[1 - \left(\omega_s - \pi/4\right)^2 / \left(2\omega_s^2\right)\right]^{-1} .$$

(2)

[9] Equation (12): I think this should be 6.11 not 0.611 if the unit of the vapor pressure is hPa (=mbar). It is correctly implemented in the code, although the wrong unit is given there (Pa instead of hPa).

Thank you for noticing. The typo in the text has been corrected.

Please note that the unit in the code [Pa] is actually correct as the expression from former equation (12) is multiplied by 100 so the result in the code really is in [Pa]. The reason for doing this is our tendency to do all the calculations in the basic SI system units.

[10] Line 227: difference in day length between what and what?

Thank you for your comment. This was obviously poorly explained in the text. The additional explanation has already been presented in the answer of comment [8].
Further graphical representation can be found in Fig. 2 of Winslow et al. (2001)

[11] Equation (22): the function of light transmission as a function of depth was somehow derived by Wu et al. based on a relationship between Secchi depth and lake depth of a range of Swedish lakes by Hakanson (1995). That means, the information of surface clarity (Secchi depth) as a function of total lake depth for a range of lakes is transferred to a function of lake clarity within a specific lake as a function of depth. In my opinion, this does not make sense. If no Secchi depth measurements or other clarity information is available for the studied lakes, I think it is preferable to use a constant default value for clarity.

We really appreciate this suggestion. We decided to adopt the arctangent model shown in Henderson-Sellers (1986) and found out that the model we previously used significantly overestimated the light extinction. As this is probably the most important process, it is not surprising that this has brought a significant improvement to the model performance. We are truly grateful and thankful for your comment!
The following has been changed/added to the text:

*The net shortwave radiation reaching a particular depth is calculated using the arctangent model from Henderson-Sellers (1986):*

$$S_n(z) = (1-\alpha)S \exp(-K_1 z)\left[1 - K_2 \tan^{-1}(K_3 z)\right],$$ (3)

*where $S_n(z)$ is the net shortwave radiation at water depth z (W m$^{-2}$), $\alpha = 0.06$ is the water surface albedo and $K_1$, $K_2$ and $K_3$ are empirical constants. $K_1$ corresponds to the light extinction coefficient $\lambda_e = 0.1$ (value of 0.1 is appropriate for clear oligotrophic lakes). $K_2$ is calculated as*

$$K_2 = 2\left[1 - (1-\beta)\exp(\lambda_e z_A)\right]/\pi,$$ (4)

*where $\beta = 0.4$ accounts for the absorption in the surface layer and $z_A = 0.6$ m is the depth of the surface absorption layer, where the exponential decay starts. The third parameter, $K_3 = 4$, is not a direct function of $\lambda_e$ and $\beta$, but it is a measure of the rapidity of falloff with depth in the upper layers.*

[12] Equations (23) and (26): I think the first epsilon is redundant in both these equations. Furthermore, reflection of the longwave radiation at the lake surface of about 3% of longwave radiation is neglected (e.g. Henderson-Sellers, 1986, https://doi.org/10.1029/RG024i003p00625). Randomly, these two things (neglecting 3% removal and adding an epsilon factor of 0.96) more or less cancel each other.

This is not a serendipitous coincidence. Generally, emissivity + reflectivity = 1, so the first epsilon in Eq. (23) and (26) actually refers to the value (1-reflectivity). Corresponding explanation has been added in the text:

*As direct measurement data of longwave radiation by pyrgeometers are not routinely available, longwave radiation may be calculated using the following formula:*

$$L_n = (1-r)L_a^\downarrow - L_s^\uparrow = \varepsilon\left[\varepsilon_a\sigma(T_a + 273.15)^4\right] - \varepsilon\sigma(T_s + 273.15)^4 , \tag{5}$$

*where r is the water reflectivity for longwave radiation, ε and $\varepsilon_a$ are the emissivities of the lake surface and the atmosphere, respectively, $T_s$ is the water surface temperature (°C), $T_a$ is the air temperature at 2 m height (°C), and σ = 5.67x10$^{-8}$ W m$^{-2}$ K$^{-4}$ is the Stefan-Boltzman constant. The emissivity of water is assumed to be 0.96 (e.g., Sun et al. 2007) and the typical relation r + ε = 1 is used in Eq. (5).*

[13] Lines 260 ff: In Crawford and Duchon (1999), f was defined as 1 minus the ratio of observed radiation to clear-sky radiation. This never reaches zero because even at 100% cloudiness, significant radiation remains. Does this have any implications for how the model is applied here?

Thank you for this comment. In the submitted version of the code we did use a simplified version where f directly equaled the cloud cover value (going from 0 to 1). However, in the new version we improved the calculation and use the cloud fraction term (clf) calculated according to Eq.(3) from Crawford and Duchon (1999). The only difference is that instead of the observed radiation we use the previously estimated value (former eq (11)). This is especially convenient as we no longer need the cloud cover as input data.
The text has been expanded accordingly:

*Assuming that the emissivity of the water droplets in the clouds is approximately equal to one, Crawford and Duchon (1999) calculate the total atmospheric emissivity as follows:*

$$\varepsilon_a = (1-f)\varepsilon_{ac} + f \tag{6}$$

*where f is the cloud fraction term defined using the ratio of the estimated surface shortwave radiation and surface clear-sky shortwave radiation:*

$$f = 1 - S_{surf}/(\tau_{cf}S_{top}), \tag{7}$$

*For clear sky conditions, the cloud fraction term equals 0. However, since the ratio of the surface solar irradiance to the clear-sky irradiance never reaches zero, the cloud fraction term never reaches the theoretical maximum of 1 even in total cloud cover conditions. Note that even though the model will be run with a time resolution of one hour, the daily mean atmospheric emissivity will be used.*

[14] Chapter 3.1.4: I don't think this approach is correct. Assume $T_{prec}$ is equal to the lake surface temperature. Then the precipitation does not change the lake surface temperature. But in the model it does increase the temperature. $T_{prec}$ should probably be replaced by ($T_{prec}$-$T_s$) in the equation?

Thank you for noticing this error which was a result of one unfortunate neglecting of a term in the expression. This has been corrected both in the code and the text.
The text has been changed as follows:

*Assuming the first lake layer in the numerical model gets completely mixed with the precipitation that falls during a time period Δt (s), then the temperature of that layer would equal:*

$$T_{1+p} = \frac{\Delta z_1 T_1 + P/(1000\times3600)\Delta t T_{prec}}{\Delta z_1 + P/(1000\times3600)\Delta t}, \tag{8}$$

*where $T_1$ and $T_{1+p}$ represent the water temperature of the first layer before and after the precipitation has been introduced in it (°C), $T_{prec}$ is the precipitation temperature (°C), $\Delta z_1$ is the thickness of the first layer (m) and P is the hourly precipitation (mm h$^{-1}$). The heat flux brought in by precipitation $H_p$ (W m$^{-2}$) can then be calculated as:*

$$H_p = \frac{1}{\Delta t}\left[\Delta z_1 + P(1000 \times 3600)\Delta t\right]\rho c_p \left(T_{1+p} - T_1\right) = \rho c_p P(1000 \times 3600)\left(T_{prec} - T_1\right). \tag{9}$$

*Since $T_{prec}$ was not available, the air temperature was used instead.*

[15] Line 320: The implicit Euler method is unconditionally stable, but it can still lead to significant errors if the time step is too large. A time step of one hour seems comparably long for this model, where the forcing data can change quite strongly from hour to hour. Did you check whether the solution would be significantly different with a shorter time step?

Unfortunately, the time resolution of our meteo data is 1h. It would definitely be interesting to check what would happen with the solution if shorter time step is used. However, in our opinion, we do not expect significant differences, due to the inertia of the system.

[16] Figure 5: There is something wrong here. The theoretical upper limit of the shortwave heat flux is the solar constant of 1368 W/m2, the typical upper limit of observed surface solar radiation is about 1000 W/m2. The July peak in the figure is 20'000 W/m2.

Thank you for noticing. A mistake was made only in the part of the code responsible for plotting the heat fluxes (instead of the mean, the monthly sum was shown). However, this was done for all the components which means that the graph was at least qualitatively correct, as well as the conclusions drawn from it. The graph has now been corrected.

[Figure]

*Figure 2 Modeled mean diurnal variations in the heat flux at the surface of Lake 12 for January (a) and July 2019 (b).*

[17] Line 396: Add some quantitative information about the error in the onset of stratification. That is difficult to read from the figures.

The following text has been added:

*For Lake 1, the position of the maximum temperature gradient in the metalimnion, between 12 and 16 m depth was captured even in the 30-day simulations (Fig. A**Error! Reference source not found.**), but the temperatures in the epilimnion are significantly overestimated in the stratification period (August) in the longer runs (Figs. **Error! Reference source not found.** and A**Error! Reference source not found.**).*
*For Lake 12, the difference between the predicted and observed position of the maximum temperature gradient is within 2 m even for the 30-day simulations, but generally it is lower. Temperature overprediction is noticed in the epilimnion, especially towards the end of the year for the simulation lengths above 10 days. The stratification began on 21 March and in the 30-day simulations it was predicted on 23 March, while the convective overturn began on 06 September while in the 30-day simulation it was predicted on 10 September.*

[18] Figures 9 and 10: for which period are these measures averaged? This should be mentioned in the caption of the figures. Also, the fact that the temperature bias at the lake surface is consistently positive even in simulations of very short duration (1 day),

seems to clearly indicate that there is something wrong with the surface heat flux parameterization (see main comment above).

Averaging periods have been added in the captions:
We agree and we do mention the surface heat flux may be one of the factors leading to the temperature bias at the lake surface. Since the major component of the net surface heat flux is the solar shortwave radiation, we did a rough check by comparing our estimations to a climatological mean for Gospić meteorological station which is the closest station for which climatological radiation data is available. Meteorological station Gospić (564 mASL) is around 40 km away from meteorological station Plitvička Jezera (579 mASL) and the region is characterized by complex orography leading to different microclimates. Furthermore we are comparing one year data to climatological data. Considering all that, no perfect match is expected. The comparison is shown in the figure below. It seems that the solar radiation was well estimated, and if anything it may be somewhat underpredicted (which is not necessary true, due to the different microclimate of the two places).

[Figure]

In the light of the model update regarding the light extinction model, it is our opinion that the main error was introduced trough this part of the code which is now improved.

[Figure]

*Figure 3 Model performance parameters for Lake 1. (calculated for all the periods with necessary data available: 6.7.-31.12.2019. and 2.7.-30.9.2020.)*

[Figure]

*Figure 4 Model performance parameters for Lake 12. (calculated for all the periods with necessary data available: 7.7.-4.11.2018; 1.1.-31.12.2019. and 2.7.-30.9.2020.)*

[19] Line 474: Again, some quantitative information on the error of the simulated onset and termination of stratification as well as the thermocline depth would be useful.

The following text has been added (Please note that this text corresponds to the new results and not the ones reported in the submitted version of the manuscript):

*First noticeable temperature rise and early beginning of stratification appear on March 21st in the observed data and March 18th in the predictions (Fig. 12). Significant temperature gradients exceeding 2 °C m-1 appear on June 12th, however in the predictions the maximum gradient appears at depth of around 2.5 m, while in the observations at depth of 5 m (Fig. 13). The thermocline depth increases during the summer and the maximum temperature gradient appears on September 21st at depth of 12 m and equals 2.5 °C m 1. On the same date the maximum predicted temperature gradient appears on the same depth but equals only 1.3 °C m 1. Actually, Fig. 13 shows that while the model accurately predicts the upper limit of the metalimnion it consistently overestimates its thickness which consequently leads to underprediction of the temperature gradient in it. December 6th may be considered as the point of complete end of stratification, with temperature gradients below 0.5 °C m-1. Although the predicted mixing depth is in agreement with the measured data, the overestimated epilimnion temperature consequently leads to temperature gradients of around 0.6 °C m 1 which persist until the end of the simulation. This is more clearly presented in Fig. 14 where the observed and modeled monthly profiles are shown. Here, it can also be seen how the model generally overpredicted the monthly mean lake temperatures. The discrepancies between the modeled and observed profiles were largest in the mixed layer during the fall and winter convective overturn. Nevertheless, the mixing depth was well captured. It is concluded that the modeled results satisfactorily reproduced the monthly mean profiles and their annual variation except after the convective overturn when higher temperature overestimation is observed.*

[20] Table 2: There are numerous lake modelling studies reporting quantitative errors compared to observed data. Below, some other studies that could be considered here, but there are many more:

- LakeMIP publications: Goyette et al. (2013), https://doi.org/10.5194/gmd-6-1337-2013
- Perroud et al. (2009), https://doi.org/10.4319/lo.2009.54.5.1574
- Read et al. (2017), https://doi.org/10.1016/j.ecolmodel.2014.07.029
- Gaudard et al. (2019), https://doi.org/10.5194/gmd-12-3955-2019
- Moore et al. (2021), https://www.sciencedirect.com/science/article/pii/S1364815221001444

I understand it would exceed the scope of this manuscript to completely review this literature, but at least the formulation that there are only few studies reporting such information should be reconsidered.

We absolutely agree that there are numerous lake modelling studies reporting quantitative errors compared to observed data. What we see as a problem and what we state in the manuscript is that there is no common systematic approach to this. Namely, different studies report different performance measures, sometimes the observation periods and measurement frequency and depths are not clearly stated or measurement are too rare to represent short term variations.

We also state that "*No performance measure data referring to certain simulation lengths were found.*" We truly weren't able to find studies calculating the performance measures in relation to the simulation period using only the end results as we do (former figures 9 and 10). The available studies mainly use all the simulation results from the beginning to the end of the run.

Thank you for the suggested literature. It certainly gave us some valuable additional information. Data reported by Read et al. (2017) and Moore et al. (2021) was be added in former Table 2.

- Stepanenko et al. (2013), https://doi.org/10.5194/gmd-6-1337-2013 (please note that the doi pointed to a Stepanenko et al. article, instead to a Goyette et al. although Goyette is one of the authors)

Stepanenko et al. (2013) do offer valuable conclusions as a model intercomparison study. However, the presented results are not directly comparably with ours. Namely, Kossenblatter is very shallow lake (average depth 2m, max depth 5m) and it is polymictic.

The analysis is run once for the summer-autumn period and as the only measure of stratification the temperature difference between the surface and 1m depth is used. They report the bias and RMSE for the surface temperature and for the temperature difference between the surface and at 1 m depth.

- Perroud et al. (2009), https://doi.org/10.4319/lo.2009.54.5.1574

Perroud et al. (2009) focus on Lake Geneva which cannot really be characterized as small. Part of Lake Geneva (Petit Lac, max depth 76m) does mix every winter, but in the deeper part (Grand Lac, max depth 309m) complete mixing occurs rarely. In the study, a period of 10 years is analyzed. The analysis is restarted every year. However, validation is based on bimonthly vertical soundings and not on continuous measurements. The RMSE is reported for couple of models.

- Read et al. (2017), https://doi.org/10.1016/j.ecolmodel.2014.07.029

Read et al. (2017) analyzed 2368 temperate lakes out of which 434 had observational data that could be used to validate model performance. However, it is not made clear for what periods and water depths is the available data.

The time period between 1979. and 2011. is analyzed with the analysis being restarted each year. The reported RMSE is calculated by using data from different lake types. It is useful that RMSE for values for all depths, for epilimnion and hypolimnion are given.

- Gaudard et al. (2019), https://doi.org/10.5194/gmd-12-3955-2019

Gaudard et al. (2019) present their online platform with almost live simulation results for 54 Swiss lakes. They do report some RMSE values however they state that the observational data is generally available on monthly bases. Again, it is not clear for what periods, what simulation lengths and what depths are these RMSE values calculated.

- Moore et al. (2021), https://www.sciencedirect.com/science/article/pii/S1364815221001444

Moore et al. (2021) present very interesting software for ensemble lake modeling which they employ for two lakes and run a one-year analysis. They report quite a few performance measures which is very useful for comparisons. However, it is not stated for which depths are these measures calculated, and what is the frequency of the observation data.

[21] Line 521: I find it surprising that the turbulent term has no effect. This would imply that for the present lakes, vertical mixing is practically exclusively driven by convection, which seems unlikely. Maybe the turbulent term is underestimated and this is the reason why the simulated thermocline position is too shallow as suggested on line 430? What are the vertical turbulent diffusivities resulting from the model?

The calculated turbulent term is generally several orders of magnitude higher than the molecular diffusion term. However, it exponentially drops with depth, so it is significant only in the first cca 1.5 m of depth. The answer to comment [7] refers to this subject.

[22] Line 542: I disagree that the position of the thermocline and its deepening were well captured. The position of the thermocline seems to be 5 to 10 m off for most of the year in Figure 13 (but see request above to provide some quantitative measures for this).

The code was updated according to some of your comments, for which we are immensely grateful, and couple of additional details we noticed. This led to major improvement of the results. Quantitative data was added as requested in the previous comments.

---

## Author Comment (AC2)

We would like to thank Referee #2 for the useful comments which helped us further improve the manuscript. Our response is as follows.

For easier referring to the each comment we also include the full Referee #2 review in black, together with our answers in blue.

The authors presented a detailed response to the comments suggesting the study would have been deeply revised. In particular, a comparison with other lake models is to be added, which would certainly add a value to the manuscript and would provide the reader with necessary information on the model performance and usability. The numerous changes described in the response imply the results differ significantly from what was presented in the original version, and the discussion assumed to be focused on the model performance compared to other lake models. After reworked in such a form, the study might provide a significant contribution to GMD and would find an appropriate readership among modelers. One remaining general question on my side is whether the proposed model has sufficient novelty compared to that of Sun et al. (2007).

Compared to the model presented in Sun et al. (2007), the present model does not neglect the turbulent diffusion for small lakes. Also we forgot to stress it uses different light attenuation approach (this will be addressed in more detail in special remarks later).

Additionally, what we consider as most important, is the fact that Sun et al. (2007), as well as number of similar papers, do not present details on determining the input data. Here we provide carefully chosen parametrizations and approximations, and incorporate them in the code itself so that the input data include only easily available meteorological variables. The text in the introduction is further refined to point this out more clearly.

*Conversely, other lake-temperature models that are forced with observational data (e.g., Bell et al., 2006; Sun et al., 2007; Martynov et al., 2010; MacKay, 2012, 2017) require both shortwave and longwave radiation component data and do not provide further details on determining them. The proposed model employs carefully chosen parameterizations of longwave and shortwave radiation. Although these parametrizations are well known, in the present study they are for the first time built into a lake temperature model, allowing the input data to include only easily available meteorological variables. Furthermore, in comparison with the model of Sun et al. (2007), the present model does not neglect the turbulent diffusion for small lakes and uses different approach for calculating the light attenuation with depth.*

Below are also remarks on the Authors' responses to the first round of comments:

[11] The shortwave radiation model of Henderson-Sellers appears to be too complex for the case when no data on the light extinction properties of the lakes are available. A one-band exponential Beer Law or the two-band Jerlov's model would provide more robust alternatives, where the value(s) of the extinction coefficient(s) might be carefully adjusted based on the comparison of the model results against observations. It is an important issue, since shortwave radiation absorption will strongly affect the final modeling results in terms of the vertical stratification as well as surface temperatures.

As suggested by Henderson-Sellers (1986), the arctangent model doesn't show significant difference to the simple model using the Beer Law as shown in the figure below (except in the surface layer).

[Figure]

Fig. 8. Depth profiles of near-surface downward irradiance in percent of surface irradiance for arctangent model and simple extinction model.

The reason for choosing the arctangent model over the Beer Law is the simplicity for implementing in a model being a single expression. Also, we believe it gives better representation of the light attenuation in the shallow layers which are usually a lot thinner than the deeper ones.
Furthermore, the arctangent model basically uses the same input as the Beer Law as the $K_1$ and $K_2$ coefficients are calculated based on the $\lambda_e$, $\beta$ and $z_A$. The only additional coefficient is the $K_3$ coefficient (a measure of the rapidity of falloff with depth within the upper layers of the water body). Suggested value $K_3=4$ is used, while we also note that the sensitivity of the light attenuation to this parameter is much lower than to the remaining parameters.
The text has been expanded to include this explanation:

*The net shortwave radiation reaching a particular depth is calculated using the arctangent model, which was chosen for its simplicity for implementation as suggested by Henderson-Sellers (1986), but also for its better representation of the light attenuation in the shallow layers which are usually a lot thinner than the deeper ones*

[12] Longwave radiation balance on the lake surface: note that r=\epsilon only in thermodynamic equilibrium, which is generally not the case for the lake surface. Better use more careful formulations here.

Thank you for this comment. The text has been corrected to:

*In Eq. 22 we assume the relation r + ε = 1, although it strictly holds only in case of thermodynamic equilibrium (which is generally not the case for the lake surface).*

---

## Author Response (AR2)

At the beginning, we would like to thank Referee #1 for useful comments which helped us improve the manuscript even further. Our response is as follows.

General comments

The manuscript has been significantly improved based on the previous reviews. However, I still think it is a bit of a mixture of topics that usually should not be combined in a single paper and much of which have relatively limited scientific novelty. But I think it is up to the journal editor to decide whether they think it is appropriate for the journal.

The manuscript in my view consists of three topics:

a) How to derive the forcing for a lake model if limited meteorological data is available (explicitly no direct observations of the radiative fluxes)? A viable way for this is now clearly described in the paper, but I don't think there is much novelty included in this approach.

b) The development of a new lake model. It remains unclear, however, which properties of the new model justify its presence besides the substantial number of already existing models. The manuscript focuses on the integration of the heat flux procedure from a), but this could have been easily integrated in any already existing lake model instead. Furthermore, due to the mixture of topics, the structure of the paper is suboptimal for a methods paper introducing a new model.

c) The numerical experiment looking at the model performance as a function of the simulation length. This is actually an interesting numerical experiment, and I am not aware that I have seen it been done elsewhere. However, it takes a disproportionate fraction of the paper, if the focus of the paper should be the introduction of a new model (as implied by the title). Furthermore, there is almost no guidance for the reader to understand the purpose and the implications of this numerical experiment. Why is this analysis performed? What hypotheses should be investigated with this approach? What general take-home messages do result from this analysis? What are the implications for the application of this and other lake models?

The leading goal of this paper was to develop a model which can be used as a black box with as little input data as possible. This can be especially useful, for example, for scientist from other fields. We do understand how the derivation of the forcing may be seen as a separate topic, but from the standpoint of creating a practical model we consider it to be a component of the model itself. Therefor we consider it is justified to unite these two topics (a and b).

Furthermore, GMD guidelines for model description papers say that "*Model description papers are comprehensive descriptions of numerical models which fall within the scope of GMD. The papers should be detailed, complete, rigorous, and accessible to a wide community of geoscientists. In addition to complete models, this type of paper may also describe model components and modules, as well as frameworks and utility tools used to build practical modelling systems, such as coupling frameworks or other software toolboxes with a geoscientific application.*" It is our opinion that our model falls into this category.

Regarding the evaluation part (c) - we took into account that it is most common, and even required, to include it in the model description paper. We didn't consider it as extensive as to require a separate paper.

We do recognize that the purpose of the said numerical experiment was not well communicated to the reader. Thank you for pointing that out. The results of this analysis are to show the model ability to provide quality short term prognosis and the rate of the result deterioration with the increase of the simulation length. The current text is now modified to address this.

In addition, the two following **major points** should be considered:

[1] The new comparison of the model with GOTM and SCHISM could be useful. However, it needs a description how the other two models were parameterized (could be in the supplement). Was exactly the same approach used for all three models (e.g., same initial profile, same meteorological forcing)? The text is a rather unclear about that.

A description of the SCHISM and GOTM model parametrization has been added in the Appendix B.

All three simulations are started with the same initial lake temperature profile. In SIMO the meteorological forcing is calculated using solely data measured at one point next to the lake. Apart from the measured air temperature and wind data (GOTM simulation) and measured air temperature (SCHISM simulation), meteorological forcing for GOTM and SCHISM was modeled with the atmospheric Weather Research and Forecasting (WRF) model (Skamarock and Klemp, 2008). As the SCHISM model is 3D (while SIMO is 1D), it requires atmospheric forcing above entire lake area (not only in a single point), thus, input data for SIMO and SCHISM could not be exactly identical. In both GOTM and SCHISM simulations, freshwater was assumed. Also, due to consistency, in both model runs the same k-ε turbulence closure scheme of Rodi (1984) was employed. Finally, both models were initialized with the lake temperatures observed at 1 January 2019 (same as SIMO).

This was already stated in the text, but it has now been slightly modified to better convey this information.

Also, I have no experience with SCHISM, but there must be clearly something wrong with the model setup. Otherwise the surface temperature couldn't possibly be systematically off by 5 °C for the entire second half of the year.

During calibration and parametrization of models, the GOTM model was set with finer vertical grid resolution, while in SCHISM the vertical resolution was coarser because it is a 3D model that consumes a lot of resources and time, and the calculations are more expensive. Likewise, the problem with SCHISM is that it covers elements in a large size range (from 1 to 200 m$^2$) and thus covers a relatively large CFL range (because it uses time implicit integration), which necessarily means that it has problems with either diffusion or numerical errors. In the end, the SCHISM model could perhaps be improved in some respects, but this would require very long time integrations (due to small elements and time steps), which was not the main goal of this work, but the development of a new model. Also in this case, vertical processes in the lake (because we are looking at long time

scales) are more important than horizontal processes (advection), which is better resolved in the GOTM model.

[2] The model does not, as far as I understand, consider the changing lake area with depth. This leads by definition to either a wrong net heat flux at the lake surface or wrong temperatures in the lake, because the ratio of surface area to volume is different in the model than in the real lake. This should somewhere be mentioned as a limitation of the model.

It would be easy to include the changing of the lake area with depth in the equations/model. However, considering that more often than not, the bathymetry of the lakes is not available, as well as our goal to keep the model as simple as possible and limit the input data, we decided to use the constant area assumption. This explanation has been added in the text (at the beginning of chapter 3):

*Considering that more often than not, the lake bathymetry is not available, as well as our goal to keep the model as simple as possible and limit the input data, it is assumed that the water body has a constant horizontal cross-sectional area (which can be of any shape).*

[3] The manuscript is rather lengthy and contains a large number of figures. It also seems occasionally repetitive. There is certainly potential to make it more concise without losing relevant information.

We are aware that the manuscript is rather lengthy, but as it encompasses all the aspects of the model plus its evaluation we find it rather hard to make it much shorter. The evaluation segment may seem a bit repetitive at first glance, as the figures for the short term sensitivity analysis and the long term simulation qualitatively resemble, however, they convey different aspects of the model performance.

**Minor comments**

[1] Line 45: I would not consider GLM (Hipsey et al., 2019) as a two-layer model

Thank you for noticing, this is absolutely true. The text has been changed to:

*Energy budget-based models assume series of well-mixed (sometimes just two, namely, the epilimnion and hypolimnion), and they use the kinetic energy produced by wind shear on the surface to account for the mixing dynamics within these two layers and/or to estimate the depths of these layers (e.g., Bell et al., 2006; Mironov et al., 2010; Hipsey et al., 2019).*

[2] Line 54: The first author's name is Råman Vinnå.

Thank you for pointing this out. The mistake has been corrected.

[3] Line 175: Eddy diffusivity is certainly not negligible in the hypolimnion of these lakes or small lakes in general. Observations in much smaller lakes typically show vertical diffusivities in the hypolimnion on the order of 10^-6 m2/s, which is one order of magnitude larger than thermal diffusivity. One of the first famous studies to investigate this is Powell and Jassby (1975, https://doi.org/10.4319/lo.1975.20.4.0530), who investigated Castle lake (0.2 km2). Other examples are Soppensee (0.23 km2; Vachon et al. 2019, https://doi.org/10.1002/lno.11172), or two small lakes (0.25 and 0.05 km2) in the Canadian Experimental Lakes Area (Quay et al., 1980; https://doi.org/10.4319/lo.1980.25.2.0201). There are certainly many more examples

available in literature. Mixing in the interior of lakes can sometimes go down to molecular levels, but basin-scale mixing almost never does.

We really appreciate the mentioned references. The text has been changed to:

*Although Sun et al. (2007) suggest that for shallow lakes (less than 50 m deep), the turbulent thermal conductivity is negligible, this is not in accordance with findings of numerous other studies which suggest that the turbulent thermal conductivity can be much larger than the molecular thermal conductivity even for shallow lakes (eg. Jassby and Powell, 1975, Quay et al., 1980. Vachon et al., 2019). It should be kept in mind that these studies often determine the turbulent diffusion coefficient based on measured change rate of lake water temperature vertical distribution, which means that the contribution of all mixing processes is included (i.e. shear-induced turbulence, breaking internal waves, boundary layer turbulence). However, the mixing processes and their contribution to turbulent mixing may differ from lake to lake. In the present study, turbulent thermal diffusion was taken into account using Eq. (3).*

[4] Line 231: The description of D is rather confusing and disagrees with that in the original paper of Winslow et al. (2001).

Maybe it was formulated a bit complicated, but the description did agree with the original paper of Winslow et al. (2001). We tried to make the text less confusing.

*The effect of cloudiness is indirectly taken into account by introducing the factor $(1-\beta rh_{Tmax})$. This is based on the finding that the solar irradiation from sunrise, when minimum humidity is expected $(rh_{Tmin}\approx 1)$, until the maximum daily air temperature (and minimum humidity $rh_{Tmax}$) is reached, is proportional to the decline of the relative humidity, $S_{surf\_Tmax}\propto(1-\beta rh_{Tmax})$. The factor $D=S_{surf}/S_{surf\_Tmax}$ is introduced to account for the surface solar irradiation from the moment when the air temperature reaches its daily maximum until sunset. D is calculated assuming that the air temperature reaches its maximum around 3pm*

[5] Line 252: add units to K1 (lambda_e) and K3. Why not just use lambda_e in the equation instead of K1, as it is used in the equation for K2 anyway?

K1 was used as we wanted to keep the equation structure as shown in Henderson-Sellers (1986).

[6] Equation (22): I don't think there is a physical reason to assume that (1 - longwave albedo r) and the emissivity of the water surface (epsilon) are identical. Why not simply keep r and eps as two variables, they can still take the values 0.04 and 0.96 to keep the results the same.

The proposed change has been adopted.

[7] Line 290: typo in "heat"

Corrected.

[8] Equations (29) and (30): should these equations not have a negative sign (positive fluxes downward)?

That is true and has been corrected. Thank you for noticing. The mistake is present only in the text and not in the code.

[9] Line 347: I still think a time step of 1 hour is rather large for the interpolation and could lead to significant numerical errors. Maybe it is ok, but it should at least be checked for one example by how much the model output changes if a smaller time step is used. I disagree with the previous reply that the available time resolution of the meteo data precludes such an analysis. The main question is whether the mixing algorithm in the

lake model produces different results for a higher time resolution and that can also be tested with meteo forcing that is constant over each hour.

The one year simulation for Lake 12 was run for dt = 60s, 600s and 3600s. Surface temperature and performance measures are shown in the figure and table below. The improvement with dt=60s compared to dt=3600s is far from dramatic. Even further reduction of the time step to dt=60s even slightly worsens the performance compared to dt=600s.

[Figure]

Comparison of the near surface water temperature for the period from 01.01.-27.12.2019. for different integration time steps

Comparison of performance measures for the period from 01.01.-27.12.2019. for different integration time steps

| Performance measure | Unit | Time step | | |
|---|---|---|---|---|
| | | 3600 s | 600 s | 60 s |
| RMSE | °C | 1,4803 | 1,4348 | 1,4352 |
| Bias | °C | 0,8504 | 0,7878 | 0,7891 |
| MAE | °C | 1,1847 | 1,1344 | 1,1375 |
| MaxAE | °C | 3,9641 | 3,9654 | 3,9654 |

The text has been modified to:

*Considering the time resolution of the available input data, the model was run with a time step of one hour (runs with finer time steps were attempted, however the performance improvements were not significant).*

[10] Line 396: Is "totally out of phase" the correct statement here?

The word "totally" has been removed.

[11] Line 405: Monomictic lakes can also mix at temperature significantly different from 4°C. It is therefore not necessary a correct assumption to initiate the model with a constant temperature of 4°C.

The temperature in the parenthesis was removed from the text. We agree that lakes don't necessarily mix at this temperature, however in this case it is true (shown by measured data).

[12] Line 457: If the reason for the too high surface temperatures was an underestimation of turbulent mixing, there should also be an underestimation of deep water temperatures for the same reason. This is clearly not the case, as the bias is positive for all depths. The reason for the bias therefore must almost certainly be the heat budget, where either some incoming heat flux is systematically overestimated or some outgoing heat flux is underestimated. Unfortunately, it is not possible to make a full heat budget calculation since the model does not consider the lake bathymetry (see point [2] above), but it could be at least approximately estimated by how much the net heat input to the lake is overestimated. For example, a mean bias of 0.5 °C (as roughly estimated from Fig. 10a) over 40 m depth would correspond to an excess heat of 84 MJ/m2. If that excess heat results over a time of 30 days, the net heat flux at the surface is overestimated by about 32 W/m2.

Indeed, but we did mention in the text that the heat flux may be overestimated which may be one of the factors leading to the estimation in the epilimnion (line 469).

One more factor that we failed to mention is the neglecting of the tributary influence. This has now been added to the text.

[13] Figure 9: add units to those metrics that have units.

Units added to the figure.

[14] Figure 15: wrong caption

Corrected to:

*Figure 1: Comparison of near surface water temperature for SIMO, GOTM and SCHISM for the period from 01.01.-27.12.2019.*

[15] Line 580: straightforward

Corrected

[16] Figures A4 and A5: captions are exchanged?

This has been corrected. Thank you for noticing.